# SSM-PixNav: State Space Models for Pixel-Guided Embodied Navigation

**Athira Krishnan R**                                        *ai22resch01001@iith.ac.in*
*Department of Artificial Intelligence*
*Indian Institute of Technology Hyderabad*

**Sumohana S Channappayya**                                  *sumohana@ee.iith.ac.in*
*Department of Electrical Engineering*
*Indian Institute of Technology Hyderabad*

**Reviewed on OpenReview:** *https://openreview.net/forum?id=RmsMd5vdBf*

## Abstract

While navigating a robot toward a specified pixel in an image, ensuring precise spatial grounding remains challenging. Existing paradigms, including object-goal, image-goal, goal-instance, and pixel-goal navigation, address this problem at different levels of granularity. We focus on pixel-goal navigation, which provides pixel-level supervision for more precise localization. Prior approaches primarily rely on RGB observations, limiting geometric awareness in scenarios where visually similar regions differ in navigability. Transformer-based policies improve temporal modeling but incur high computational cost, and there is a lack of standardized open benchmarks for reproducible evaluation. We address these limitations in three ways. First, we propose RGBD-PixNav, which integrates depth directly into the policy. Second, we introduce a Mamba-based State Space Model (SSM) for efficient temporal modeling, along with causal SSM policies and a depth-gating mechanism for adaptive fusion of RGB and depth features. Third, we construct the PixNav Trajectories dataset using HM3D scenes in Habitat-Sim to enable a reproducible benchmark. Experiments show that the Causal SSM-RGB and RGBD variants outperform strong baselines, improving the success rate by approximately 0.4 while reducing the parameter count to $\approx 27M$ (half). The models also demonstrate robustness to observation noise and varying history lengths. Code and dataset are available at https://github.com/lfovia/SSM-PixNav.

## 1 Introduction

Robotics is an interdisciplinary field of engineering that focuses on building agents to make human life safer and more efficient. Robot navigation is the study of successfully guiding an agent from point A to point B. In the classical approach, navigation is handled through hand-engineered components, including the path planner, Simultaneous Localization and Mapping (SLAM), and the control loop to track the agent. With the emergence of AI, research on robot navigation takes two roads: one towards mapping the environment with multimodal perceptions Huang et al. (2023b), Huang et al. (2023a), and the other towards predicting the optimal waypoint or action Zhao et al. (2025), Zhang et al. (2024), Zhang et al. (2025) directly from observations to achieve a goal. These models are either vision-only or vision-language-based action prediction models.

Embodied navigation adopts an end-to-end learning paradigm that enables agents to directly map raw observations to actions, thereby making them more adaptable to novel, unstructured, and dynamic environments. These agents can be deployed as a cobot, assistive agents Niklasson et al. (2024), etc. Recent works Cai et al. (2024), Zhao et al. (2025) leverage the power of foundation models Liu et al. (2024), Kirillov et al. (2023) and VLM's Li et al. (2022), OpenAI (2023), Beyer et al. (2024), Zhu et al. (2023) for semantic awareness

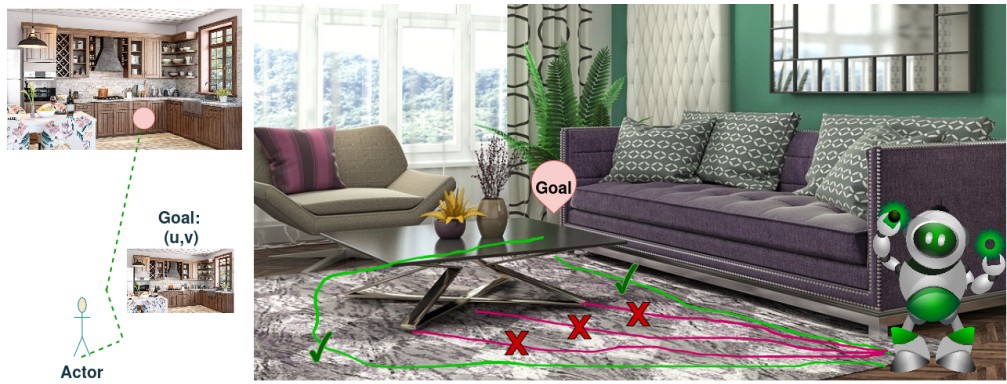

Figure 1: A simple illustration of PixNav. Given an image and a goal coordinate, the agent predicts waypoints to reach the target. In the example, the robot navigates toward a location near the potted plant; the green path indicates a successful trajectory, while the red path results in a collision.

and navigation. These embodied navigation policies are designed to be lightweight and later deployed to a real robot. For a realistic environment, simulators began to rely on photorealistic datasets, such as HM3D Ramakrishnan et al. (2021) and MP3D Chang et al. (2017). These datasets cover 3D indoor scenes from an apartment. Habitat-sim Savva et al. (2019), Szot et al. (2021), Puig et al. (2023) provide highly realistic scene rendering at several thousand frames per second for indoor scenes in the datasets listed above.

When an agent is instructed to perform object goal navigation, it will move to any instance of the object (closest to the agent) in the environment. So we cannot ensure the agent will navigate to a specific instance itself. To address this, image-goal navigation uses a goal image. Here, the agent tries to reach a similar image scene, but it is unaware of the most important element needed to ensure a match. Through pixel-goal navigation, an agent is given a goal image and a 2D mask to determine which pixels to focus on, as shown in Figure 1. This ensures the agent can reach a location more precisely compared to other methods. Because the goal is not defined by a goal category type or coordinates, agents can navigate toward arbitrary visual targets, including dynamic or previously unseen objects. This work considers pixel-goal navigation as the main task and object-goal navigation as a downstream task.

Pixel navigation policies typically learn to map sequences of observations and goal masks directly to low-level actions, effectively bypassing explicit mapping or pose tracking modules. Existing works on pixel navigation rely on datasets that are not publicly available. Also, valuable depth information that is already available is not being leveraged. Further existing architectures are transformer-based, making them less suitable for deployment on edge devices. The key contributions to pixel navigation in this work are listed below,

1. RGBD-PixNav: An RGBD-based pixel navigation policy with transformers, a direct extension of RGB-PixNav that leverages RGB images and depth information to improve scene understanding and action prediction.

2. Causal RGB-SSM PixNav: A light-weight causal RGB-based pixel navigation policy with Mamba, that relies on RGB images only to improve scene understanding and action prediction.

3. Causal RGBD-SSM PixNav: An extension of the Causal RGB-SSM PixNav model that relies on RGB images and depth maps.

4. PixNav Trajectories, an open-source dataset of trajectories curated from the HM3D environment using Habitat's oracle agent Shortest Path Follower (SPF), addressing the lack of public supervised data for pixel navigation.

5. Robustness analysis is performed by perturbing the observations with a mild Gaussian noise.

6. Comprehensive empirical analysis shows that SSM-based policies outperform transformer-based baselines across both navigation success and computational efficiency.

Table 1: Model performance on entire validation fold of proposed PixNav Trajectories dataset. * is marked for weights from PixNav Cai et al. (2024).

| Model | Total loss | Action loss | Distance loss | Tracking loss | Action Accuracy |
|---|---|---|---|---|---|
| PixNav* | 4.54 | 4.51 | 0.31 | 0.0005 | 16.81 |
| PixNav (our baseline) | 0.94 | 0.93 | 0.02 | 0.0005 | 65.41 |

Through these contributions, we demonstrate that the SSM-PixNav model is an effective natural fit for embodied navigation, thereby enabling the development of efficient, generalizable and reproducible agents.

## 2 Related Works

Image-goal navigation Majumdar et al. (2022) leverages the CLIP model Radford et al. (2021) and ResNet features to encode the goal and the current observation. In Majumdar et al. (2022), policy is trained through reinforcement learning, where the agent tries to learn through exploration and exploitation of its own understanding of the environment.

In prior work Cai et al. (2024), a pixel navigation policy was proposed through imitation learning. The agent relied on RGB observations and goal descriptors. Goal descriptors include a goal image and a binary mask to show the region of interest. In this model, depth information was utilized solely to identify the goal location via pinhole camera projection. Krishnan R et al. (2026) attempted to establish a depth-only model for pixel navigation. Through decision-level fusion of the logits from depth and RGB pixel navigation models, Krishnan R et al. (2026) have shown that the ensemble model outperformed the RGB variant. While recent work has shown promising results in pixel-based navigation using transformer-style policies, reproducibility and fair benchmarking remain open challenges.

To address these issues, we curate trajectories from the HM3D dataset Ramakrishnan et al. (2021) and release a balanced version of the same, referred to as the PixNav Trajectories dataset. This dataset includes diverse trajectories generated by a SPF oracle agent with a turn radius of 0.5, promoting reproducibility and fair comparison of pixel navigation algorithms. Even though we tried to follow the steps shared in PixNav Cai et al. (2024) closely, the evaluation results of model weights from Cai et al. (2024) on the PixNav Trajectories dataset were subpar. Given this, we have established a baseline for the same model by fine-tuning only the three MLP heads to predict action, steps to goal and goal coordinate. The evaluation performance of both models is compared in Table 1. The validation losses and accuracy metrics indicate that our PixNav Trajectories dataset differs from the one used by the authors of Cai et al. (2024).

## 3 Proposed Models for Pixel-Guided Navigation

### 3.1 Problem statement

Given a set of trajectories with observations, a goal descriptor, and a set of allowed actions, we aim to predict the next action, the goal coordinate, and the steps to the goal. Goal descriptors include an RGB image and a pixel coordinate. Trajectory length is limited to a maximum step count of 64 due to computational constraints. The agent is allowed to take 6 valid actions: {stop, forward, left, right, look up, look down}. Learning is conducted in a supervised setting using the proposed PixNav Trajectories dataset. We can further formulate this as a multi-head classification (action) and regression (coordinate and steps to goal) problem.

### 3.2 Transformer-based Models for Pixel-guided Navigation

Transformers Vaswani et al. (2017), unlike RNNs and LSTMs, can model long-range dependencies in data through self-attention. For an image token, we compute Query $(Q) \rightarrow$ what am I looking for?, Key $(K) \rightarrow$ what do I contain? and Value $(V) \rightarrow$ what information do I carry? We can compute this attention score as,

$$\text{Attention}(\mathbf{Q}, \mathbf{K}, \mathbf{V}) = \text{Softmax}(\frac{\mathbf{Q}\mathbf{K}^T}{\sqrt{d}})\mathbf{V}, \tag{1}$$

where $d$ is the number of features in each key/query vector. Thus, now with attention, each token becomes a context-aware representation. In multihead attention (MHA), multiple attention heads are processed in parallel. In PixNav, they use transformer decoder layers with masked multihead self-attention, thereby preventing the agent from peeking into future frames. From $n$ attention heads, $head_1, ..., head_n$,

$$\text{MHA}(\mathbf{Q}, \mathbf{K}, \mathbf{V}) = \text{Concat}(head_1, ..., head_n)\mathbf{W}, \tag{2}$$

### 3.2.1 RGB-PixNav

This section describes our approach to reproducing the results from RGB-PixNav Cai et al. (2024) using our proposed PixNav Trajectories dataset. In RGB-PixNav, the agent solely relies on RGB goal descriptors and observations to predict the next action. The agent is exposed to dataset $D = \{(x_1, y_1), (x_2, y_2), ....(x_n, y_n)\}$ where the sample $x_i = (goal\ image; goal\ mask; observation\ history)$ and the label $y_i = (action_i, distance_i, mask_i)$. Here, $(x_i, y_i)$ corresponds to the data tuple of step $i$ in a trajectory. In pixel navigation, the agent only sees pixel-level details as input. A goal mask is a binary 2D mask generated from the goal image that indicates the target's pixel coordinates. The ground-truth pixel is expanded to an $8 \times 8$ local neighborhood centered on the target pixel, with all pixels within the window labeled as positive. The goal image and goal mask together serve as the goal descriptors. The agent also receives RGB observations from the environment at each time step. As mentioned in Cai et al. (2024), the goal descriptors are concatenated along the channel dimension and passed through the goal encoder network (4-channel ResNet-18 He et al. (2016)). The resultant embedding is average-pooled and projected to sizes 768 and 256 via linear layers as the goal input token and the goal concat token, respectively. The past $K$ observations serve as a history to help the agent learn temporal representations from the data. For this, observation data is also passed through an observation encoder (ResNet-18) network. The resultant embedding is pooled and projected to 512 dimensions. The so-obtained features are passed through 4 layers of a transformer decoder with masked attention to ensure causality. The resultant embedding is a 768-dimensional vector and is passed to three MLP heads one each for action prediction, coordinate prediction, and distance estimation. The action prediction head ensures the model learns to predict actions closely like an Oracle agent (ground truth). The coordinate prediction head ensures the agent can keep track of the goal pixels during motion. The distance prediction head ensures the agent estimates the distance to the target. The architecture shown in Figure 2 represents the proposed RGBD PixNav policy. The same architecture without depth observation inputs can be referred to as the RGB-PixNav policy. Cai et al. (2024) proposed a multiobjective loss function for pixel navigation with three components to address three heads. Action loss $\mathcal{L}_{act}$ can be formally defined as the cross-entropy loss between the predicted actions $p_{ij}$ and the corresponding ground truth $y_{ij}$ for a sample of $K$ steps.

$$\mathcal{L}_{\text{act}} = -\frac{1}{N} \sum_{i=1}^{N} \sum_{j=1}^{C} y_{ij} \log(p_{ij}). \tag{3}$$

Distance loss, $\mathcal{L}_{dist}$, is the Mean Squared Error (MSE) between the predicted and actual distances over $K$ timesteps.

$$\mathcal{L}_{\text{dist}} = \frac{1}{K} \sum_{i=1}^{K} \left(\hat{d}_i - d_i\right)^2, \tag{4}$$

where $\hat{d}_i$ is the predicted distance at step $i$ and $d_i$ is the ground truth distance at step $i$. This equation computes the average squared error between the predicted and actual distances across all $K$ steps. Similarly, tracking loss, $\mathcal{L}_{track}$, is the Mean Squared Error (MSE) between the mask constructed from the predicted coordinates and the ground truth goal map over $K$ timesteps.

$$L_{\text{track}} = \frac{1}{K} \sum_{i=1}^{K} \left(\hat{\mathcal{M}}_i - \mathcal{M}_i\right)^2, \tag{5}$$

where $\hat{\mathcal{M}}_i$ is the mask reconstructed with predicted coordinates at step $i$ and $\mathcal{M}_i$ is the ground truth (actual) mask at step $i$. The overall objective is formulated as in (6), to minimize the weighted combination of each loss terms (3)-(5),

$$\theta^* = \arg \min_{\theta} \mathbb{E}_{(x_i, y_i) \sim D} \left[\mathcal{L}_{\text{act}}(x_i, y_i) + 0.1 \cdot \mathcal{L}_{\text{dist}}(x_i, y_i) + \mathcal{L}_{\text{track}}(x_i, y_i)\right]. \tag{6}$$

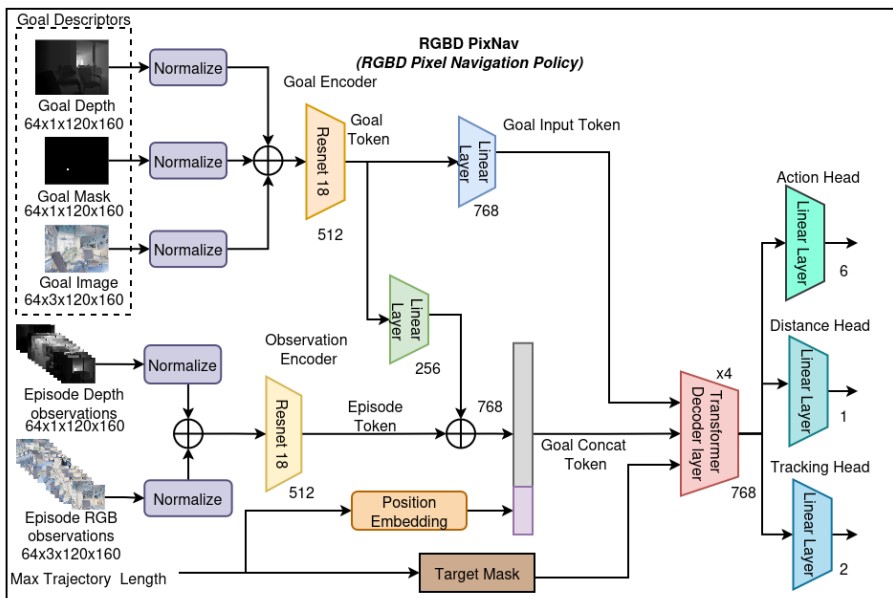

Figure 2: **Proposed architecture of RGBD-PixNav policy**. This architecture is a direct extension of Cai et al. (2024), here to introduce depth, the RGB and depth observations are concatenated along the channel dimension to goal and episode observation. In the case of RGB PixNav, episode depth observations are not considered.

### 3.2.2   RGBD-PixNav

In RGBD-PixNav, a direct extension of RGB variant, our experiment to understand the value of depth modality. Depth data is introduced to describe the goal and, at the observation level, to predict an action. The agent uses the dataset $D = \{(x_1, y_1), (x_2, y_2), ....(x_N, y_N)\}$ where the sample has $x_i = (goal\ image; goal\ mask; \textbf{goal depth map}; RGB\ observation\ history; \textbf{depth observation history})$ and $y_i = (action_i, distance_i, mask_i)$. The goal mask is computed from the goal image as in 3.2.1. The goal descriptors are concatenated along the channel dimension and passed through the goal encoder network (5-channel ResNet-18 He et al. (2016)); the resultant embedding is average-pooled to size 512 and projected to sizes 768 and 256 via linear layers. At each time step, the agent receives RGB and depth observations from the environment. Goal descriptors, a history of $K$ RGB images, and $K$ depth map observations are passed to the model as a sample. The RGBD observation data is passed through the observation encoder (a 4-channel ResNet-18). Observation embeddings are concatenated and projected to 512 dimensions. The goal token and fused observation feature with position embedding are passed through 4 layers of a transformer decoder to obtain the resultant vector, which is passed to 3 MLP heads for action prediction, coordinate prediction, and distance estimation. The same multiobjective loss function in section (3.2.1) with three components, corresponding to three heads, is reused here. The model objective can also be formulated as (6), as in RGB PixNav, where the dataset now contains RGBD samples.

### 3.3   Mamba-based Modeling for Pixel-guided Navigation

Mamba Gu et al. (2024) is a lightweight, selective state space model (SSM) that operates with linear time complexity $O(T)$ and is specifically designed for sequence modeling. They have a strong inductive bias toward temporal modeling, making them a natural fit for navigation tasks. Vanilla SSMs, on the other hand, are input agnostic. A discrete-time linear SSM can be expressed as shown in (7).

$$s_t = \mathbf{A}s_{t-1} + \mathbf{B}u_t \ , \ y_t = \mathbf{C}s_t, \tag{7}$$

where variable $s_t \in \mathbb{R}^N$ is the hidden state (memory), $u_t \in \mathbb{R}^D$ is the input and $y_t \in \mathbb{R}^D$ is the output. The matrices $\mathbf{A} \in \mathbb{R}^{N \times N}$ is the state transition matrix (memory decay), $\mathbf{B} \in \mathbb{R}^{N \times D}$ is the input projection

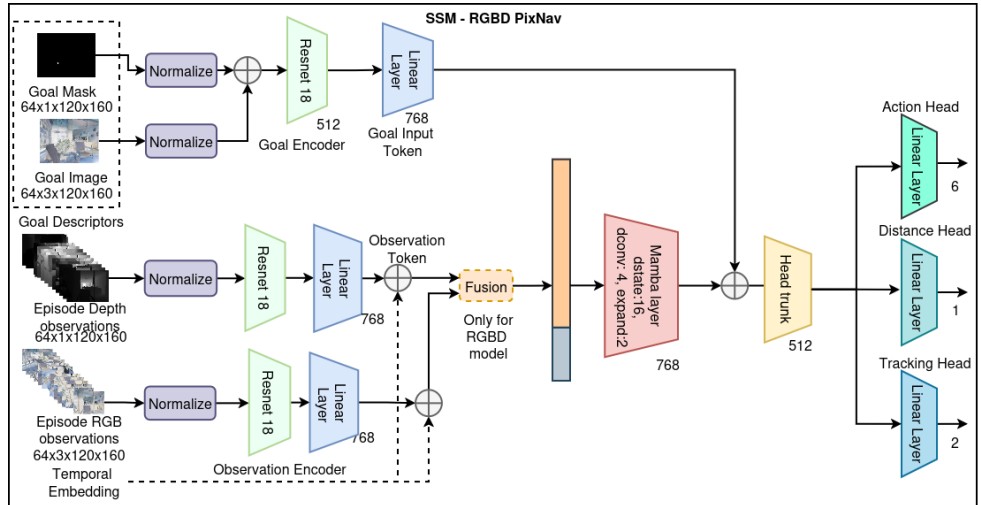

Figure 3: **Proposed architecture of SSM-based Pixel-guided Navigation**. The model is learned end-to-end; the features from the goal descriptors and each modality's observations are encoded and fused, after which the temporal features are learned using a Mamba layer. Temporal embedding introduced ensures the ordering of the observation tokens.

matrix, and $\mathbf{C} \, \epsilon \, \mathbb{R}^{D \times N}$ is the readout matrix. $N$, $D$ represent the state size and token embedding dimension, respectively. In SSM, with these matrices fixed, every input is treated equally. In Mamba, fixed dynamics are replaced with input-conditioned dynamics as shown in (8).

$$s_t = \mathbf{A}(\boldsymbol{x}_t)s_{t-1} + \mathbf{B}(\boldsymbol{x}_t)\boldsymbol{x}_t \ , \ \boldsymbol{y}_t = \mathbf{C}(\boldsymbol{x}_t)s_t, \tag{8}$$

where $\boldsymbol{x}_t$ is the observation embedding and matrices $\mathbf{A}(\boldsymbol{x}_t), \mathbf{B}(\boldsymbol{x}_t), \mathbf{C}(\boldsymbol{x}_t)$ are learned functions of input. This, in turn, enables selective information propagation. In our proposed approaches, Mamba layers effectively capture sequential dependencies in observations through state-space modelling, unlike the transformer.

### 3.3.1   SSM-RGB PixNav Model

In this section, we propose a PixNav model trained with Mamba SSM layers. The proposed architecture is depicted in Figure 3 without depth inputs. In SSM-RGB PixNav, an embodied pixel navigation agent is built solely on the RGB modality. So, during training, the agent uses the dataset $D$ with samples as defined in section 3.2.1. Goal image and goal mask, generated as in 3.2.1, serve as the goal descriptors. The agent also receives RGB observations from the environment at each time step. Goal descriptors and a history of $K$ RGB image observations are passed to the model. The goal descriptors are then projected to a goal token of dimension 768 through a linear layer to match the Mamba layer dimension. The observation data is also passed through the observation encoder (ResNet 18) network. The embeddings from each frame are pooled and projected to 384 dimensions. These observation tokens are passed to the Mamba layer. The encoders help learn a better spatial representation, whereas the Mamba layers help with temporal modeling. The result from the Mamba layer is concatenated to the Goal token. The features are passed through a head trunk layer composed of a sequential layer with layernorm, a linear layer, ReLU, and dropout with a rate of 0.1. The resultant head-trunk feature of 512 dimensions is passed through 3 heads to predict the action, the coordinates, and the distance to the goal.

Temporal embedding ensures temporal ordering, but to ensure strict causality, a causal variant of the policy is proposed as shown in Figure 4. It introduces a mask that is 0 for future time steps and 1 for current and prior time steps. With the new mask, the input to the model now becomes $x_i = (goal\ image; goal\ mask; observation\ history;\ mask)$. Thus, by multiplying the mask with the history tokens, Mamba is ensured to see only the data up to $T$ timesteps. Here, we combine the goal token before passing it to the Mamba layer. Representation from the Mamba layer is passed through the head trunk layer and 3 prediction heads. This variant of the model is referred to as Causal SSM-RGB PixNav.

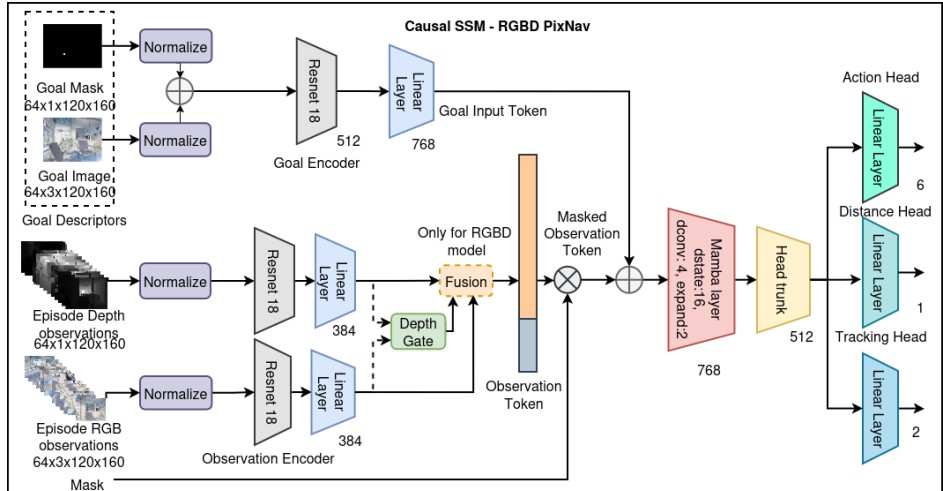

Figure 4: **Proposed architecture of Causal SSM-RGBD PixNav with depth gate**. In the case of Causal SSM-RGBD, Pixnav learned observation representations from the RGB and depth encoders, which are fused by concatenation. The fused embedding is multiplied by the mask and combined with the goal tokens. The goal-conditioned temporal features from the Mamba layer are passed through the trunk layer and the MLP heads. Whereas in Causal SSM-RGBD PixNav with depth-gate, an additional depth gate is added to learn the mixing ratio between the RGB and Depth features.

### 3.3.2 SSM-RGBD PixNav Model

Given the success of the depth extension of RGB PixNav, extending SSM-RGB PixNav with a depth modality is a natural next step. In SSM-RGBD PixNav, an embodied pixel-guided navigation agent is built using RGB and depth modality inputs, leveraging the proposed architecture, as depicted in Figure 3. The agent uses the dataset $D$ described in section (3.2.2). Goal image and goal mask, as in 3.2.1, serve as the goal descriptors. The $K$ RGB and depth observations act as history to help the agent understand temporal cues. The observation data is passed through modality specific observation encoders (a 3-channel ResNet-18 for RGB and a 1-channel ResNet-18 for Depth). The resultant embedding from each encoder is projected to 384 dimensions. The same is separately embedded in time for ordering, then fused by concatenation. This fused feature is passed through Mamba layers to get a 768-dimensional vector. The resultant embedding is combined with the goal token and passed through the head trunk, yielding a 512-dimensional feature vector. The resultant vector is passed to 3 MLP heads for action prediction, coordinate prediction, and distance estimation. The same multiobjective loss function from (3.2.1) with three components, corresponding to three heads, is used.

For a causal variant, a mask is introduced at the input level as in Causal SSM-RGB Pixnav shown in Figure 4 and mixed with the fused observation token before being passed to the Mamba layer. This ensures the Mamba attends only to observations from previous time steps. In this variant of the RGBD model, features are fused via concatenation, allowing the agent to consider depth data even when it is noisy. To control this dependence on depth at every time step, a depth gate is introduced to learn when to rely on depth data, by analyzing observation data as shown in Figure 4. The learned mixing ratio is scaled to 50% to ensure the RGB is the dominant feature. With the learned mixing ratio, the representations are mixed and later masked for causality. The fused representation is combined with a goal token for goal conditioning and later passed through the Mamba layer. The agent now understands when to look at depth data and which fraction is important. With depth gating, masking, and early goal conditioning, the agent can make better predictions.

$$token_{depth} = 0.5 * \text{Sigmoid}(\text{Linear}(\text{Concat}(token_{rgb}, token_{depth}))) * token_{depth} \tag{9}$$

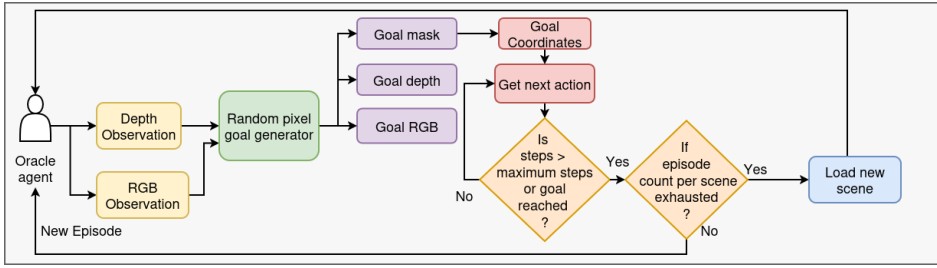

Figure 5: **Dataset generation pipeline** used to generate PixNav Trajectories dataset.

## 4 PixNav Trajectories Dataset

The dataset created for training pixel-guided navigation through imitation learning, as part of Cai et al. (2024), was not publicly released. To address this lacuna, following the descriptions in the prior work closely, we have curated a dataset of navigation trajectories generated by the SPF oracle agent. Dataset curation and rollout validation of trained pixel-guided navigation policies were done in Habitat-sim Savva et al. (2019), Szot et al. (2021), Puig et al. (2023). Dataset splits are curated with $K = 64$, from the train and evaluation folds of HM3D dataset. During data generation in an episode, we first try to identify feasible random goal points in the scene, as shown in Figure 5. To this end, the depth data is projected to a point cloud format and tied with RGB pixels. A point is now randomly selected to ensure it is reachable. By doing so, we introduce randomness in the data itself. Despite numerous efforts, we were unable to reproduce the results reported in prior work due to random noise in the data. Following a supervised setting, the SPF (oracle agent) was used in the Habitat-sim to generate the ground truth. While collecting the data, the agent is initialized with the following settings: maximum step count of 64, a camera field of view of 79 degrees, and a camera height of 0.88m. The agent is made to navigate through 300,000 episodes. The agent skips the episode if the selected goal points proposed are not valid in an episode. If a valid goal is identified, the agent records it as a dataset sample. As depth data is freely available for an agent in Habitat-sim using the HM3D dataset, we have also curated depth maps for the RGBD model in the PixNav Trajectories dataset. In a trajectory, we have a goal described by an RGB goal image, a binary goal mask, and a depth map. Corresponding to each step in the trajectory, we have an RGB observation image, a binary goal mask tracked by the oracle agent, and a depth map, with remaining steps recorded as distances and the action taken by the oracle agent. Table 2 lists the different difficulty levels' definitions considered in the trajectories covered in the dataset. Policy rollout validation is done online. We relied on the SPF agent to obtain ground truth data, which achieved a success rate of 91.87%, 93.10% and 57.14% on easy, medium and hard episodes respectively on the rollout evaluation on HM3D dataset with Oracle Success Rate as a metric of dataset quality Wang et al. (2019). The left column of inputs in Figure 2 shows a sample from proposed dataset.

Table 2: Distribution of train and validation splits by difficulty level.

| Difficulty Level | Train Sample | Val Sample | Definition |
|---|---|---|---|
| Easy | 34555 | 3445 | Distance to goal is less than 3m. |
| Medium | 33673 | 3430 | Distance to goal is between 3 and 5m. |
| Hard | 32016 | 3058 | Distance to goal is between 5 and 8m. |

## 5 Results

### 5.1 Pixel-Guided Navigation Rollout Validation

During training, after the model converges, we perform rollout validation on a few epochs from the region of convergence. During rollout validation, the agent is randomly initialized in the scene, and a random pixel goal is selected via point cloud projection and feasibility checks. The agent is now tested for its capability to reach the goal within the maximum allowed steps. As the trained agents move according to their own predictions, they start to deviate from the oracle agent's path, and the error begins to accumulate. This

Table 3: Pixel-guided navigation policy evaluation results in the HM3D datasets. *This weight is shared by the authors of Cai et al. (2024) and is trained on a private dataset. All training was done with image resolution $120 \times 160$ following the training setup shared by Cai et al. (2024). All hyperparameters are detailed in Appendix A.1.

| Model | Image Size | Model Size | Easy SR (↑) | Easy SPL (↑) | Medium SR (↑) | Medium SPL (↑) | Hard SR (↑) | Hard SPL (↑) |
|---|---|---|---|---|---|---|---|---|
| RGB PixNav *Cai et al. (2024) | $224 \times 224$ | 54.449M | 0.7497 | 0.7238 | 0.3448 | 0.3339 | 0.0476 | 0.0476 |
| RGB PixNav *Cai et al. (2024) | $120 \times 160$ | 54.449M | 0.3123 | 0.2925 | 0.0345 | 0.0345 | 0.0476 | 0.0476 |
| Ensemble (RGB + Depth) Krishnan R et al. (2026) | $120 \times 160$ | 109.39M | 0.4610 | 0.4304 | 0.0625 | 0.0625 | 0.1818 | 0.1818 |
| RGB PixNav (baseline) | $120 \times 160$ | 54.449M | 0.3676 | 0.3441 | 0.1739 | 0.1726 | 0.1429 | 0.1429 |
| SSM-RGB PixNav | $120 \times 160$ | 27.624M | 0.7545 | 0.7417 | 0.3091 | 0.3049 | 0.2000 | 0.2000 |
| Causal SSM-RGB PixNav | $120 \times 160$ | **27.314M** | **0.8043** | **0.7905** | **0.4808** | **0.4727** | **0.2273** | **0.2273** |
| RGBD PixNav | $120 \times 160$ | 54.455M | 0.5361 | 0.4970 | 0.3800 | 0.3571 | 0.1500 | 0.1500 |
| SSM-RGBD PixNav with Mamba | $120 \times 160$ | 38.534M | 0.5736 | 0.5350 | 0.3111 | 0.2987 | 0.2000 | 0.1980 |
| Causal SSM-RGBD PixNav | $120 \times 160$ | 38.780M | 0.6894 | 0.6670 | 0.3200 | 0.3160 | **0.3000** | **0.3000** |
| Causal SSM-RGBD PixNav with depth gate | $120 \times 160$ | **38.485M** | **0.7187** | **0.6934** | **0.4615** | **0.4520** | 0.2727 | 0.2727 |

error accumulates over time as the agent tries to reach the goal. The trained policies are validated on HM3D Ramakrishnan et al. (2021) via rollout validation over 1000 episodes. Success Rate (SR) and Success Weighted Path Length (SPL) are the metrics identified to report the performance. A traversal is considered successful if and only if the distance to the goal at the end is $\leq$ 1m. The results obtained are tabulated in Table 3. From the numbers, we can see that the Causal SSM-RGB PixNav outperforms all the other models, and the Causal SSM-RGBD PixNav with a depth gate follows. More details on the need for causal variant and the role of depth gate are further explained in A.9 and A.10, respectively. The proposed models with the Mamba layer achieve remarkable gains in navigation metrics.

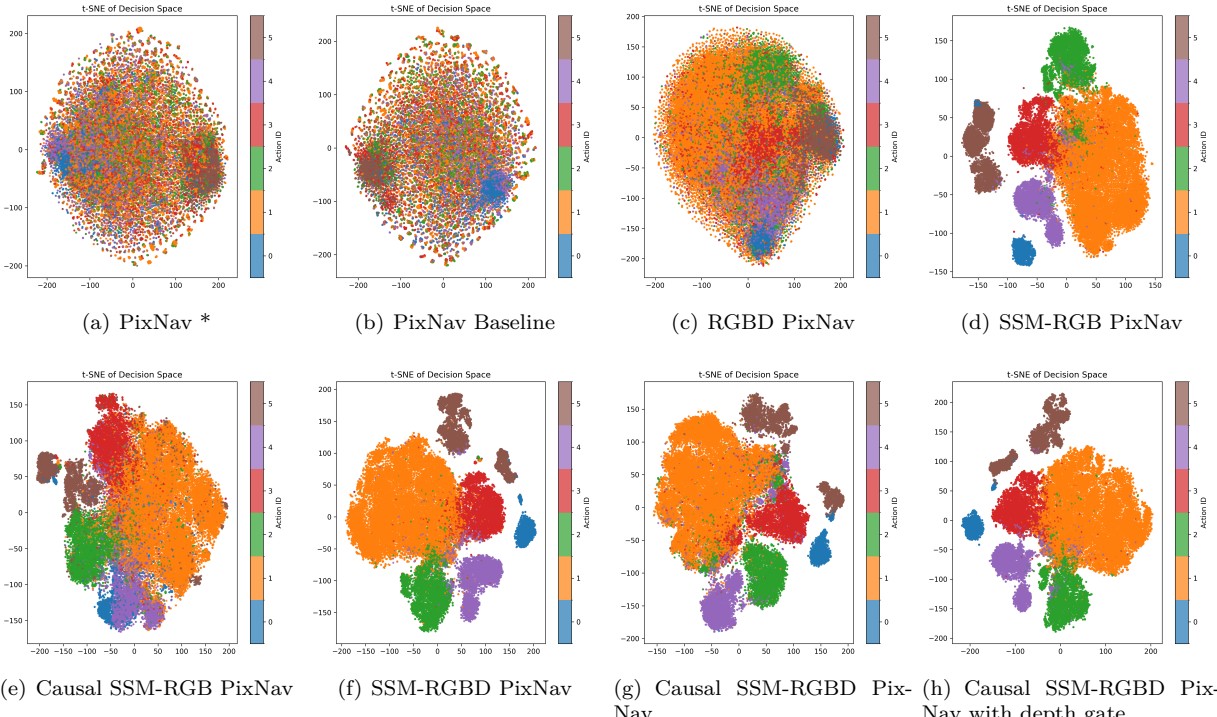

(a) PixNav *   (b) PixNav Baseline   (c) RGBD PixNav   (d) SSM-RGB PixNav

(e) Causal SSM-RGB PixNav   (f) SSM-RGBD PixNav   (g) Causal SSM-RGBD PixNav   (h) Causal SSM-RGBD PixNav with depth gate

Figure 6: **t-SNE plots of the representations learned** by all trained models from the validation trajectories in the PixNav Trajectories dataset. Subfigure (c) shows that introducing depth improves decision-making, whereas (d)-(h) have better separation, showing that Mamba is able to learn the goal-conditioned dynamics better. Labels 0 to 5 represent the 6 valid actions.* indicates weights from prior work Cai et al. (2024)

t-SNE plots are a means to visualize the separability of the learned representations. Through t-SNE, high-dimensional features can be projected to a lower-dimensional space in a supervised manner. To assess the models' ability to separate actions, t-SNE plots are generated from the validation fold data in the PixNav Trajectories dataset. The plots show better feature separability with Mamba-based models, which validate the numbers in Table 3. We attribute this to the Mamba model's ability to better capture data dynamics. As an ablation, a variant of the Causal SSM-RGB PixNav model was trained without the Mamba layers and with LSTM to learn the contribution of the Mamba layers. Results are discussed in A.5. Tracking capability is an important factor that helps the agent navigate to a pixel goal. The tracking performance of the best RGB and RGBD models during rollout evaluation from different time stamps is depicted in Figure 7. In Causal SSM-RGB PixNav, the goal in the next room is more closely tracked, whereas in Causal SSM-RGBD PixNav with depth gate, the agent at times gets confused by similar regions in the frame.

Since contributions to pixel goal navigation are limited, we compare our proposed models to those of the closely related task of goal instance navigation. Metrics reported in Table 9 from the HM3D dataset show that our proposed SSM-based policy achieves a higher SPL, indicating near-optimal trajectories in successful cases.

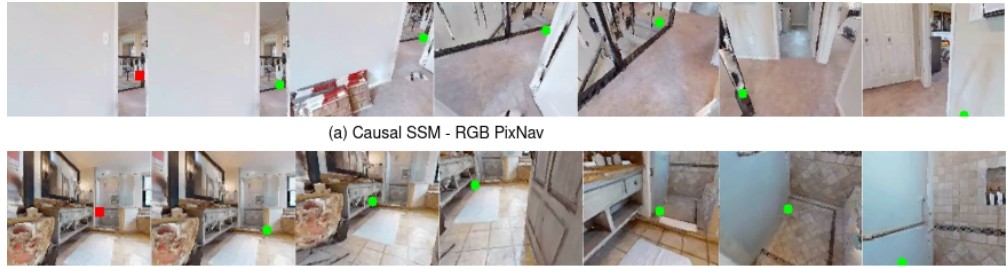

(a) Causal SSM - RGB PixNav

(b) Causal SSM - RGBD PixNav with depth gate

Figure 7: **Goal pixel tracking capability** of (a) Causal SSM-RGB PixNav, (b) Causal SSM-RGBD PixNav with depth gate is shown for random pixel goals in HM3D scenes. Pixel goal (red) and predicted goal coordinate (green) during rollout validation for episode images at different timesteps. It is clear that the Causal SSM-RGB PixNav model tracks the goal pixel more consistently. As depth gate regulates geometric cues, tracking relies primarily on appearance-based feature continuity. Additionally, temporal feature modulation introduced by depth gating might have slightly affected the stability of the representations passed to the Mamba layer.

## 5.2 Cross Dataset Validation

This experiment evaluates the generalization capability of pixel-guided navigation policies trained on the proposed PixNav Trajectories dataset, which is curated from the HM3D dataset. The dataset selected for

Table 4: Cross-dataset rollout evaluation of Pixel goal navigation policy on MP3D dataset.

| Model | Image Size | Easy SR (↑) | Easy SPL (↑) | Medium SR (↑) | Medium SPL (↑) | Hard SR (↑) | Hard SPL (↑) |
|---|---|---|---|---|---|---|---|
| RGB PixNav (baseline) | $120 \times 160$ | 0.3542 | 0.3335 | 0.0510 | 0.0505 | 0.0909 | 0.0909 |
| SSM-RGB PixNav | $120 \times 160$ | **0.6431** | **0.6327** | .1735 | 0.1713 | **0.1212** | **0.1212** |
| Causal SSM-RGB PixNav | $120 \times 160$ | 0.6418 | 0.6277 | **0.2128** | **0.2090** | 0.0968 | 0.0968 |
| RGBD PixNav | $120 \times 160$ | 0.3595 | 0.3400 | 0.1020 | 0.0979 | 0.0606 | 0.0606 |
| SSM-RGBD PixNav | $120 \times 160$ | 0.4234 | 0.3975 | 0.2121 | 0.2030 | 0.0667 | 0.0667 |
| Causal SSM-RGBD PixNav | $120 \times 160$ | 0.6117 | **0.5964** | **0.2292** | **0.2244** | **0.1667** | **0.1667** |
| Causal SSM-RGBD PixNav with depth gate | $120 \times 160$ | **0.6183** | 0.5963 | 0.2021 | 0.1992 | 0.0645 | 0.0645 |

cross-dataset evaluation is the MP3D dataset Chang et al. (2017). The rollout evaluation is also conducted with the same camera settings across 1000 episodes from the validation fold of the MP3D dataset. The numbers from Table 4 depict that the models trained on HM3D scenes are able to generalize to pixel-guided navigation problems in MP3D scenes. Mamba-based models still exhibit improved performance over the baseline, as seen in Table 3.

### 5.3 Robustness to Perturbation

In practice, all robots are prone to slight stability issues while navigating, which, in turn, affects the quality of the captured image frame and, eventually, the quality of the inferred features. To this end, an experiment was conducted on the trained model by perturbing the RGB and depth observation with additive Gaussian noise of varying standard deviation. The results obtained from all PixNav models for this experiment are tabulated in Table 5. The RGB PixNav model is the least robust to observation noise. As the noise increases, we can see the metrics drop significantly, especially in medium and hard cases. With SSM-based models, we can see that they perform consistently and the metrics degrade more gracefully. In RGBD variants, depth serves as a geometric prior; however, when the depth signal is noisy, the model becomes more sensitive to perturbations, as expected, due to degraded geometric cues. Causal variants with higher performance provide stronger temporal filtering and greater noise resilience.

Table 5: Robustness of proposed PixNav policies towards Gaussian Noise of mean zero.

| Model | Standard Deviation | Easy | | Medium | | Hard | |
|---|---|---|---|---|---|---|---|
| | | SR (↑) | SPL (↑) | SR (↑) | SPL (↑) | SR (↑) | SPL (↑) |
| PixNav* | 0.02 | 0.1005 | 0.0853 | 0.0000 | 0.0000 | 0.0500 | 0.0500 |
| (Author's weights | 0.03 | 0.0916 | 0.0774 | 0.0192 | 0.0163 | 0.1364 | 0.1364 |
| from Cai et al. (2024)) | 0.04 | 0.1038 | 0.0839 | 0.0189 | 0.0189 | 0.0800 | 0.0800 |
| | 0.05 | 0.1110 | 0.0945 | 0.0000 | 0.0000 | 0.0769 | 0.0769 |
| RGB-PixNav | 0.02 | 0.3643 | 0.3398 | 0.1500 | 0.1478 | 0.1176 | 0.1176 |
| (baseline) | 0.03 | 0.3501 | 0.3288 | 0.0833 | 0.0833 | 0.2400 | 0.2400 |
| | 0.04 | 0.3470 | 0.3273 | 0.1250 | 0.1224 | 0.1579 | 0.1579 |
| | 0.05 | 0.3530 | 0.3326 | 0.0784 | 0.0784 | 0.2500 | 0.2500 |
| SSM-RGB PixNav | 0.02 | 0.7459 | 0.7350 | 0.4681 | 0.4661 | 0.1739 | 0.1739 |
| | 0.03 | 0.7076 | 0.6982 | 0.2500 | 0.2481 | 0.0000 | 0.0000 |
| | 0.04 | 0.6843 | 0.6750 | 0.2807 | 0.2803 | 0.0000 | 0.0000 |
| | 0.05 | 0.6507 | 0.6391 | 0.3000 | 0.3000 | 0.0870 | 0.0870 |
| Causal SSM-RGB PixNav | 0.02 | **0.7596** | **0.7466** | **0.3600** | **0.3505** | **0.1852** | **0.1852** |
| | 0.03 | 0.7595 | 0.7442 | 0.2727 | 0.2671 | 0.0952 | 0.0952 |
| | 0.05 | 0.7051 | 0.6909 | 0.2679 | 0.2624 | 0.1562 | 0.1562 |
| RGBD-PixNav | 0.02 | 0.2674 | 0.2297 | 0.1026 | 0.1026 | 0.0000 | 0.0000 |
| | 0.03 | 0.3084 | 0.2536 | 0.0357 | 0.0357 | 0.0000 | 0.0000 |
| | 0.04 | 0.2331 | 0.1941 | 0.0000 | 0.0000 | 0.0000 | 0.0000 |
| | 0.05 | 0.2072 | 0.1804 | 0.0851 | 0.0851 | 0.0000 | 0.0000 |
| SSM-RGBD PixNav | 0.02 | 0.4213 | 0.3838 | 0.1702 | 0.1629 | 0.1200 | 0.1200 |
| | 0.03 | 0.3939 | 0.3624 | 0.1509 | 0.1509 | 0.0455 | 0.0455 |
| | 0.04 | 0.4262 | 0.3863 | 0.1667 | 0.1572 | .0833 | 0.0833 |
| | 0.05 | 0.4246 | 0.3874 | 0.1702 | 0.1682 | 0.0400 | 0.0400 |
| Causal SSM-RGBD PixNav | 0.02 | 0.6750 | 0.6519 | 0.1875 | 0.1806 | **0.1538** | **0.1538** |
| | 0.03 | 0.6446 | 0.6254 | 0.2830 | 0.2762 | 0.2222 | 0.2222 |
| | 0.04 | 0.6317 | 0.6129 | 0.2364 | 0.2348 | 0.2609 | 0.2609 |
| | 0.05 | 0.6178 | 0.5988 | 0.1136 | 0.1136 | 0.1739 | 0.1739 |
| Causal SSM-RGBD PixNav | 0.02 | **0.7092** | **0.6860** | **0.2500** | **0.2413** | 0.0690 | 0.0690 |
| with Depth gate | 0.03 | 0.6494 | 0.6323 | 0.2444 | 0.2356 | 0.1579 | 0.1579 |
| | 0.04 | 0.6093 | 0.5901 | 0.3061 | 0.2996 | 0.0625 | 0.0625 |
| | 0.05 | 0.6071 | 0.5888 | 0.3191 | 0.3148 | 0.1429 | 0.1429 |

## 6 Discussion

In this work, we first aim to establish a reproducible baseline for PixNav using only the RGB modality. We then added depth modality as a clear extension by concatenating depth data along the channel dimension. This simple step led to a large gain in the success rate of $\approx 0.1685$ on the easy episodes. The behaviour is consistent across episodes of varying complexity and datasets. However, the computational complexity of these models is of $O(N^2)$, and therefore, we looked for a lighter variant of the solution. In the SSM-based models, Mamba layers were incorporated to capture the dynamics in the observation features. We proposed a lighter model $\approx 27M$ with a significant performance gain of $\approx 0.39$ on the easy episodes. Temporal ordering is already handled, but to ensure causality, a mask was introduced, boosting performance by $\approx 0.44$ on easy episodes relative to the baseline. Training setup and hypermeter used for each variant is briefed in A.1

**Why could Mamba have helped?** To understand this part more closely, at each time step $t$, latent state $\boldsymbol{h}_t$ can be defined in terms of input-dependent state matrices $\mathbf{A}$ and $\mathbf{B}$ as,

$$\boldsymbol{h}_t = \bar{\mathbf{A}}_t \boldsymbol{h}_{t-1} + \bar{\mathbf{B}}_t \boldsymbol{x}_t, \tag{10}$$

where $\bar{\mathbf{A}}_t$ is the transition matrix and $\bar{\mathbf{B}}_t$ is the input projection matrix. Unrolling the recurrence gives,

$$\boldsymbol{h}_t = \sum_{k=1}^{t} \left( \prod_{j=k+1}^{t} \bar{\mathbf{A}}_j \right) \bar{\mathbf{B}}_k \boldsymbol{x}_k, \tag{11}$$

which shows that the latent state aggregates past observations through a sequence of learned, time-varying transition operators. Goal conditioning is introduced as an additive latent bias $\boldsymbol{g}$, resulting in,

$$\hat{\boldsymbol{h}}_t = \boldsymbol{h}_t + \boldsymbol{g} = \sum_{k=1}^{t} \mathbf{K}_{t,k} \boldsymbol{x}_k + \boldsymbol{g}, \tag{12}$$

where $\mathbf{K}_{t,k} = (\prod_{j=k+1}^{t} \bar{\mathbf{A}}_j)\bar{\mathbf{B}}_k$. This formulation allows the policy to reason about observations relative to the target in a goal-shifted latent space, without explicitly modifying the underlying dynamics. Although SSM-RGB PixNav performs slightly better overall, depth information offers additional geometric cues that improved the performance of RGBD variant in difficult scenarios. As an ablation study, we have varied $K$, the number of history frames during rollout evaluation, to see how many history frames the model actually leverages. More details of the experiment can be seen in A.3. The results show that higher $K$ yields better temporal features. Similarly, an experiment was conducted to evaluate the model's zero-shot object goal

Table 6: **Sensitivity Analysis.** Results from rollout evaluation of agent on HM3D dataset by varying $K$, the number of history frames.

| Model | $K$ | Easy | | Medium | | Hard | |
|---|---|---|---|---|---|---|---|
| | | SR (↑) | SPL (↑) | SR (↑) | SPL (↑) | SR (↑) | SPL (↑) |
| Causal SSM-RGB PixNav | 4 | 0.3625 | 0.3322 | 0.0727 | 0.0697 | 0.2000 | 0.2000 |
| | 8 | 0.5982 | 0.5739 | 0.3061 | 0.2968 | 0.0455 | 0.0455 |
| | 16 | 0.7957 | 0.7834 | 0.2195 | 0.2091 | 0.1053 | 0.1053 |
| | 32 | **0.8043** | **0.7906** | **0.5000** | **0.4920** | **0.2273** | **0.2273** |
| Causal SSM-RGBD PixNav with gate | 4 | 0.4023 | 0.3749 | 0.0909 | 0.0857 | 0.0000 | 0.0000 |
| | 8 | 0.5497 | 0.5241 | 0.2449 | 0.2285 | 0.0909 | 0.0909 |
| | 16 | 0.7049 | 0.6833 | 0.2439 | 0.2318 | 0.1579 | 0.1579 |
| | 32 | **0.7187** | **0.6934** | **0.4615** | **0.4520** | **0.2727** | **0.2727** |

navigation capability, leveraging guidance from an open-source VLM. The details of the same are given in (A.7). The results show that the SSM models perform slightly better than the author's weights Cai et al. (2024). While testing the models on NVIDIA AGX Orin with the OAK-D Pro camera, it was noted that the mean time to predict an action for SSM models is reduced by 10-15ms compared to the transformer-based variants. SSM variants show higher speed-up in inferencing on edge compute, with the Causal SSM-RGB PixNav model reaching 17%, which is crucial for mobile agents. More details of this experiment can be found in (A.8).

## 7 Conclusions

This work addresses the pixel-guided navigation problem, traditionally approached using only RGB inputs in embodied navigation, by extending the modeling framework to incorporate available depth information via RGBD-PixNav. The incorporation of depth maps allows the agent to leverage additional geometric cues, enhancing navigation performance in visually complex environments. To address the lack of an open-source dataset for supervised training of pixel navigation policies, a new benchmark dataset was curated for this effort. Toward better temporal modelling, Mamba layers were introduced. With changes in the goal-fusion strategy, the proposed SSM-based models were developed to clarify what to focus on.

Experimental results demonstrate that the proposed models, Causal SSM RGB PixNav followed by Causal SSM RGBD PixNav with depth gate, consistently outperform the RGB-only baseline and other proposed policy variants in pixel navigation tasks. The proposed policies also exhibit robust behaviour to mild Gaussian noise and variations in history frames. Proposed Causal SSM-RGB PixNav achieved a speed up in inference time of 17% on edge compute. The current approach relies on supervised learning, and the agent is exposed predominantly to clean expert data. Future directions include incorporating Dagger-based fine-tuning or other strategies to improve policy robustness in more realistic scenarios and addressing the sim-to-real gap.

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

# A Appendix

## A.1 Training setup

The models are trained on the proposed dataset. For training and evaluation, 100000 trajectories from the train fold and 10000 trajectories from the validation fold were used. All models were trained for 30 epochs. All fine-grained training details like learning rate, optimizer, scheduler are mentioned in Table 7 for reproducibility. The images were resized to 120x160 resolution and normalized. All transformer model training was done on a 64GB Mac Studio, and Mamba model training was done on a NVIDIA 5090. All rollout evaluations were conducted on an NVIDIA L40 machine.

## A.2 Metrics

The efficiency of any model needs to be compared, but to compare them, we need some measuring aids. In DL, we term them as metrics. The metrics generally used for navigation domains Sun et al. (2024) include Success Rate, Success Weighted Path Length, Soft SPL, and Distance to Goal. Each metric and its formulation are briefly described in the subsections below.

Table 7: Training set up used for the suite of pixel-guided navigation policies.

| Model | Optimizer | Scheduler | Learning rate | Weight Decay |
|---|---|---|---|---|
| RGB PixNav | Adam | LR Scheduler | 1e-5 | 1e-4 |
| SSM - RGB PixNav | Adam | CosineAnnealingLR | 1e-4 | 3e-4 |
| Causal SSM - RGB PixNav with Mamba Causal | AdamW | ReduceLROnPlateau | 1e-4 | 1e-4 |
| RGBD PixNav | Adam | LR Scheduler | 1e-5 | 1e-4 |
| SSM - RGBD PixNav | AdamW | CosineAnnealingLR | 7e-5 | 1e-4 |
| Causal SSM - RGBD PixNav | AdamW | CosineAnnealingLR | 7e-5 | 1e-4 |
| Causal SSM - RGBD PixNav with depth gate | AdamW | CosineAnnealingLR | 7e-5 | 1e-4 |

### A.2.1 Success Rate

Success Rate (SR) is the percentage of successful attempts of navigation an agent made to reach the goal. An agent is considered successful if it reaches within a goal radius of 1m. So mathematically the same can be framed as,

$$SR \ = \ \frac{1}{N} \sum_{i=1}^{N} \begin{cases} 1, & \text{if success} \\ 0, & \text{else}, \end{cases} \tag{13}$$

where N is the number of episodes considered. As a successful agent aimed for, a higher SR is preferred.

### A.2.2 Success Weighted Path Length

For a navigation agent, success is important, but the effort or path length taken to reach a goal is equally important. The longer the path length, the more energy will be required to navigate it. So, through Success Weighted Path Length (SPL) Anderson et al. (2018), a trade-off is established to weight successes and path length. Out of N episodes considered, in each episode, $l_i$ is the length of the shortest/optimal path from the starting position of the agent to reach the goal. Similarly, $p_i$ is the length of the path actually taken by the agent. Let $S_i$ be the success, which takes the value 1 for success and 0 for failure. Now SPL can be mathematically defined as,

$$SPL \ = \ \frac{1}{N} \sum_{i=1}^{N} S_i \frac{l_i}{\max(l_i, p_i)}. \tag{14}$$

SPL takes a value of 0 if success is 0; otherwise, its value ranges from 0 to 1. A higher SPL is preferred because it means a shorter path length to the goal.

### A.2.3 Soft SPL

Soft SPL (SSPL) is a softer version of SPL, which does not depend on the success factor. So, even if the attempt is unsuccessful, we can still implement this measure. For the same setting considered while defining SPL, SSPL can be defined as,

$$SPL \ = \ \frac{1}{N} \sum_{i=1}^{N} \frac{l_i}{\max(l_i, p_i)}. \tag{15}$$

### A.2.4 Distance to Goal

Distance to goal (DTG) is measured once the agent issues a STOP action/maximum allowed number of steps is exhausted. In this stage, the geodesic distance from the agent's current position to the goal location is measured. As reaching closer to the goal increases the likelihood of success, a lower DTG is preferred.

### A.3 Varying the number of history frames

Mamba has a selective-scanning property, so the model examines unique frames. The longer the history, the better the temporal insights the model learns. But how to validate that the higher the number of history frames considered, the better the navigation?
To this end, an ablation was performed at rollout validation by varying/ truncating the history observations

to different ranges, $K = \{4, 8, 16, 32\}$. The experiment was conducted on the best-performing models from each RGB and RGBD variant to assess changes in performance. The results are tabulated in Table 6, which shows that as $K$ decreases, knowledge about the environment is lost/truncated, leading to the representations learned. This directly affects the performance. The same trend is visible in both the variants tested, which revalidates that the higher the number of history frames considered, the better the temporal representations learned by the model. Another observation is that, in both cases, the performance with 32 frames is similar to that with 64 frames. This shows that the Mamba layer, through selective scan, attends to 32 frames even when we use 64 frames.

## A.4 Robustness to Pertubation

The observation frames are perturbed with Gaussian noise. The mean of the noise is kept zero, but the standard deviation is varied from 2 to 5 percent. A sample observation image with different levels of noise injection is shown in Figure 8.

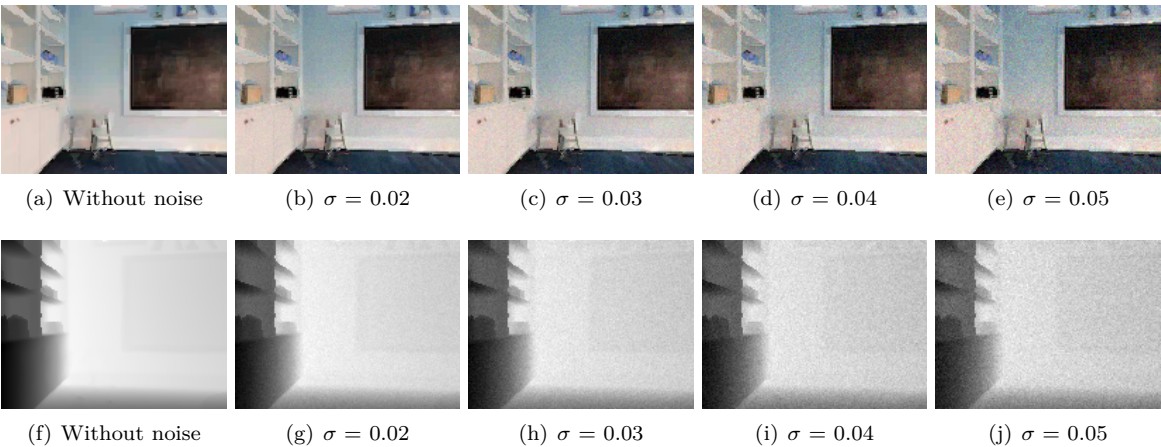

(a) Without noise    (b) $\sigma = 0.02$    (c) $\sigma = 0.03$    (d) $\sigma = 0.04$    (e) $\sigma = 0.05$

(f) Without noise    (g) $\sigma = 0.02$    (h) $\sigma = 0.03$    (i) $\sigma = 0.04$    (j) $\sigma = 0.05$

Figure 8: **Noise augmented episode observation**. Sample episode RGB observation (a) and its noisy versions (b)-(e) from a Gaussian of different standard deviations. Similarly depth observation (f) and its noisy versions in (g)-(j) from a Gaussian of different standard deviations.

## A.5 Effect of Mamba Layer

To assess the contribution of the Mamba layer in the proposed Causal SSM - RGB PixNav model, a variant was trained and evaluated without it. This model, with 22.952M parameters, takes the masked version of the fused encoded observation features and concatenates them with goal features. Here, because the model is less complex, the goal and observation features are projected to 384 dimensions. Rollout performance of the same is recorded in Table 8. Even the t-SNE plot shown in Figure 9 clearly depicts the lack of a Mamba-like efficient temporal modeling.

As an ablation, we have also tried replacing the Mamba layer in Causal SSM-RGB PixNav with an LSTM layer. After training under a similar setup, the policy was evaluated. The rollout evaluation metrics from Table 8 show that this 24.14M-parameter policy performs well on easy episodes ($\approx 3\% \downarrow$ compared to Causal SSM-RGB PixNav). But the performance drops substantially as the difficulty increases.

While Transformers can encode temporal context in observation tokens, they still rely on cross-attention for direct goal–observation matching, which benefits from the presence of geometric (depth) information in both. In contrast, SSMs compress temporal information into a latent state that accumulates geometry from observations. As a result, RGB goal descriptors are sufficient, and depth is only required in observation encoding. From training and evaluating a Causal SSM-RGBD PixNav with a depth gate and an RGBD goal descriptor, we see that the metrics and TSNE plots confirm our assumptions.

Table 8: Pixel navigation policy evaluation results in the HM3D datasets. The model used here is the same as the proposed Causal SSM-RGB PixNav, but Mamba layers are skipped.

| Model | Dataset | Easy | | Medium | | Hard | |
|---|---|---|---|---|---|---|---|
| | | SR (↑) | SPL (↑) | SR (↑) | SPL (↑) | SR (↑) | SPL (↑) |
| Causal SSM-RGB PixNav | HM3D | **0.8043** | **0.7905** | **0.4808** | **0.4727** | **0.2273** | **0.2273** |
| | MP3D | 0.6418 | 0.6277 | **0.2128** | **0.2090** | **0.0968** | **0.0968** |
| Causal SSM-RGB PixNav | HM3D | 0.2468 | 0.2373 | 0.0727 | 0.0727 | 0.1000 | 0.1000 |
| (w/o Mamba) | MP3D | 0.0824 | 0.0807 | 0.0000 | 0.0000 | 0.0303 | 0.0303 |
| Causal LSTM-RGB PixNav | HM3D | 0.7764 | 0.7685 | 0.2826 | 0.2781 | 0.1500 | 0.1451 |
| | MP3D | **0.6598** | **0.6484** | 0.1700 | 0.1696 | 0.0800 | 0.0800 |
| Causal SSM-RGBD PixNav | HM3D | 0.7764 | 0.7685 | 0.2826 | 0.2781 | 0.1500 | 0.1451 |
| with depth gate | MP3D | 0.6183 | 0.5963 | 0.2021 | 0.1992 | 0.0645 | 0.0645 |
| Causal SSM-RGBD PixNav with | HM3D | 0.4242 | 0.4161 | 0.0545 | 0.0545 | 0.1111 | 0.1111 |
| depth gate and RGBD goal descriptor | MP3D | 0.4902 | 0.4798 | 0.0306 | 0.0306 | 0.0606 | 0.0606 |

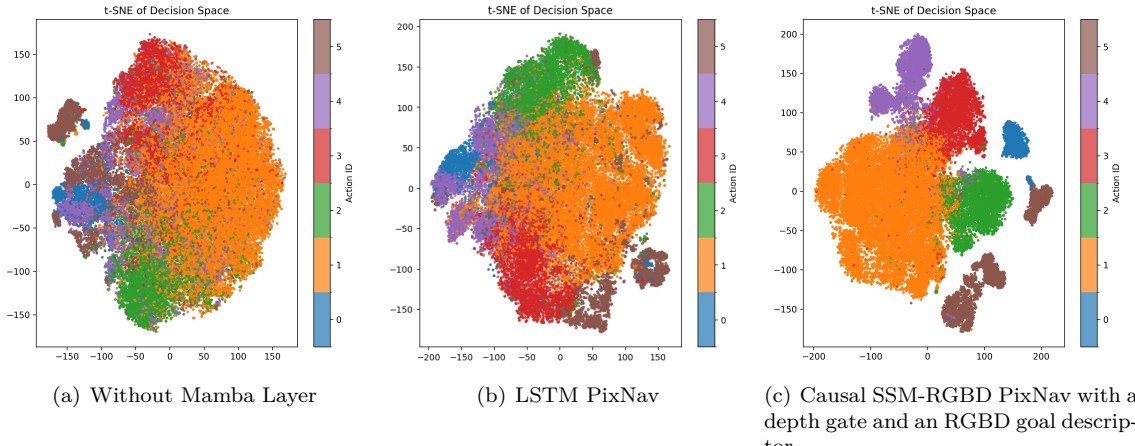

(a) Without Mamba Layer

(b) LSTM PixNav

(c) Causal SSM-RGBD PixNav with a depth gate and an RGBD goal descriptor

Figure 9: **t-SNE plot for SSM-RGB PixNav model without Mamba layer and Causal LSTM-RGB PixNav model**. In the model without the Mamba layer, we observe that spatial features help to some extent, but feature separability is lower than that of the proposed SSM variants shown in Figure 6. LSTM variant shows better separability than one without Mamba layer, but not to the level of Causal SSM-RGB PixNav shown in Figure 6.

## A.6 Goal Instance Navigation

In goal instance navigation, given a goal image and its goal category type, the agent is expected to move towards the goal. Even though in pixel navigation we do not consider the object type, but consider the pixel location, this is the most similar problem in research.

Table 9: **Metric from SOTA Goal Instance navigation models** on HM3D dataset from Yin et al. (2025). * indicates the metrics were taken from Yin et al. (2025) as the pre-trained weights or scripts to reproduce were missing. Metrics of Yin et al. (2025) are reported from 200 episodes from the HM3D dataset.

| Task | Model | SR | SPL |
|---|---|---|---|
| Goal Instance navigation | OVRL-v2-IIN* | 0.248 | 0.118 |
| Goal Instance navigation | GOAT* | 0.374 | 0.161 |
| Goal Instance navigation | Unigoal | **0.68500** | 0.25050 |
| Pixel Navigation | Causal SSM-RGB PixNav | 0.5041 | **0.4968** |
| Pixel Navigation | Causal SSM-RGBD PixNav with depth gate | 0.4843 | 0.4727 |

From the metrics, we can see that our lightweight standalone models achieve a competitive success rate compared to Unigoal Yin et al. (2025), which relies on VLMs to make decisions. But the most important observation is the substantially higher SPL in our trajectories. This ensures that our model can plan efficient trajectories, especially during successful detours.

## A.7 Zero-shot object goal navigation

This section aims to test the model's suitability for solving object-goal navigation. As the model lacks the capability to understand object type/text guidance, we propose a method that utilizes the foundation models Grounding DINO Liu et al. (2024) and SAM Kirillov et al. (2023). Here, the agent is initialized in an episode scene in Habitat using the HM3D dataset. For the object goal described in the dataset, the agent tries to complete object goal navigation.

Given an object goal, the agent first tries to determine the probable direction to explore to find the specified object. For example, a bed is found in the bedroom, a sink is found in the kitchen, and so on. To this end, the agent rotates by $360^o$ and captures the observations. Considering the FOV as $90^o$, and there are 12 steps to reduce the overlap, every alternate observation is omitted to construct a panorama. The panorama is passed to a VLM to determine the probable direction. Now the agent turns in that direction and passes the observation to the object detector to detect the object in the frame. If the object is present, the current observation image serves as the goal image, and the SAM-generated mask serves as the goal mask for the pixel navigation agent. If the object is absent, the agent looks for the floor class. The observation and the SAM-generated mask are now passed to the pixel navigation agent. If the agent is stuck, it tries to capture the panorama and plan again. This loop of actions continues until the agent reaches the goal. An attempt is considered successful, iff the agent reaches within 1m proximity to the goal.

It is evident that performance has dropped with the BLIP planner. If the planner predicts the wrong direction, the agent tries to look for the object or floor category. It will go by floor category if the object is

Table 10: **Zero-shot object goal navigation** results from Pix Nav models on 100 HM3D validation episodes.

| Model | Resolution | SR (↑) | SPL (↑) | SSPL (↑) | Distance To Goal (↓) | Distance To Predicted Goal (↓) |
|---|---|---|---|---|---|---|
| PixNav* | $120 \times 160$ | 9 | 0.0517 | 0.0961 | 5.894 | 1.3238 |
| PixNav(baseline) | $120 \times 160$ | 2 | 0.02 | 0.0730 | 5.585 | 0.8667 |
| SSM-RGB PixNav | $120 \times 160$ | **9** | **0.0638** | **0.1129** | **5.826** | 1.1713 |
| Causal SSM-RGB PixNav | $120 \times 160$ | 3 | 0.0233 | 0.0937 | 5.560 | 1.0742 |
| RGBD PixNav | $120 \times 160$ | 0 | 0 | 0.0011 | 5.627 | 0.8387 |
| SSM-RGBD PixNav | $120 \times 160$ | 0 | 0 | 0.0079 | 5.538 | 0.8202 |
| Causal SSM-RGBD PixNav | $120 \times 160$ | 0 | 0 | 0.0024 | 5.547 | 0.8202 |
| Causal SSM-RGBD PixNav with depth gate | $120 \times 160$ | 0 | 0 | 0.0061 | 5.566 | 0.8205 |

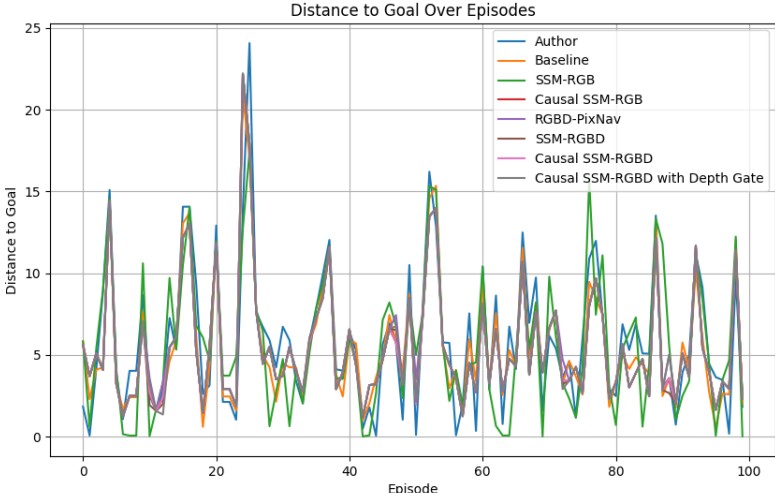

Figure 10: **DTG measures the closeness of the goal position to the stopping position**. The DTG of different models is shown here.

not visible. Being indoors, some floor space will be visible, which will mislead the agent in the wrong direction. This is expected to be a problem with our open-source VLM-guided zero-shot object goal navigation. But when we analyze the distance to the goal closely, we can see that even though our model is not able to make it a success, the agent is, on average, able to reach the goal more closely and consistently with the proposed models. The comparison of the distance to goal metric for different models is shown in Figure 10. In this section we use object goal navigation annotations on HM3D dataset, from numbers its clear that the object looked as goal by our PixNav model is probably another instance of goal. The detections being validated by DINO its another instance, during the trial the agent tries to reduce distance to this goal. Our agent donot explore the first instance it detects is taken as the goal for PixNav model. This shows indicates that depth module enabled the agent to look for another instance of goal than the annotated one. As the distance to predicted goal or goal passed to proposed PixNav model is $\leq 1m$ they can also be considered a success.

### A.8 Performance comparison on edge device

To evaluate the performance of the trained pixel navigation policies on a real robot with power and compute constraints, a performance comparison was conducted on an edge device. The target device used is NVIDIA AGX Orin with the OAK-D Pro sensor. To facilitate real-time testing, a random goal pixel is used to compute the goal mask within each image frame. Now the goal descriptors and the agent's observations at each time step are passed to the trained policy to make a prediction. Maximum steps were limited to 64. For each model, the average time taken to predict an action is tabulated in Table 11. The results show that SSM models with approximately half the model size can predict an action with reduced latency. Given the mean inference time taken for the baseline model, $t_b$ and proposed model $t_p$, we compute the speed-up percentage as follows,

$$\text{Speed-up} = \frac{t_b - t_p}{t_b} \qquad (16)$$

The variation in the time to predict an action at each step in a trajectory is depicted in Figure 11. Through careful modeling that balances reduced model size with improved performance, we find that our proposed models perform with low latency on our compute-constrained target device.

Results show that the mean inference time is reduced by $\approx$10-15s with the proposed SSM variants. Additionally, these policies achieve a maximum speed-up of 17% for SSM-RGB PixNav, which is very crucial for a mobile agent. In this section, we have evaluated and compared inference time primarily on an edge

Table 11: **Performance comparison on edge device**. On the selected edge compute and sensor, the models were executed, and the average inference time from a trajectory is reported.

| Model | Average step-level inference time (s) | Speed-up |
|---|---|---|
| Baseline | 0.0847 | - |
| Causal SSM-RGB PixNav | **0.0703** | **0.1700** |
| Causal SSM-RGBD PixNav with depth gate | 0.0770 | 0.0909 |

compute to determine the latency. But in the future, we plan to extend the work to test on real hardware by addressing sim-to-real challenges, including fine-tuning the policy to account for sensor noise, recovery mechanisms, and control.

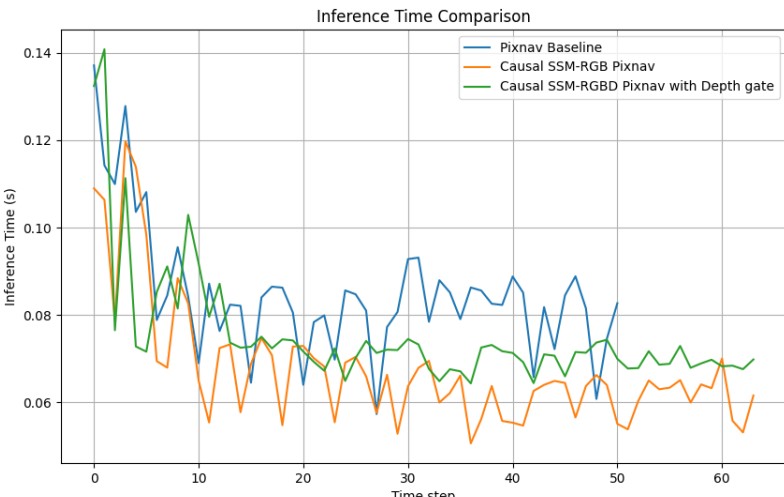

Figure 11: **Inference time comparison**. Time taken to predict an action in all steps taken by an agent in a trajectory with different policies.

### A.9   Need for causal Mamba variant

Mamba is causal in training sequential data processing. During training, we pass all $K$ observations and jointly calculate the loss. Because the data is not processed sequentially during training, there can be data leakage. During evaluation, execution is sequential. This can adversely affect the model's generalization during evaluation. To address this issue we introduce a mask to ensure the future timestep are masked.

### A.10   Effect of Depth gate

The depth maps offers geometric awareness to the agent. By introducing depth in Causal SSM-RGBD PixNav the performance is not near to RGB variant. In SSM models as we donot guide when to look at depth data unlike transformers with attention, the depth data confused the model. To this end a depth gate was introduced to learn when to leverage depth. By learning the depth gate parameter, we try to make the model learn, how much to rely on depth. Through this step the depth gated variant bags $\approx 3\%$ improvement in SR. For a trajectory from HM3D during rollout evaluation for each time step, the observations, learned depth gate mixing ratio and predicted goal coordinates are shown in Figure 12. We visualize the learned depth gating values across timesteps for a representative trajectory. The model assigns low gate values when RGB observations are sufficient (t=1–2), while higher values are observed in cluttered or geometrically

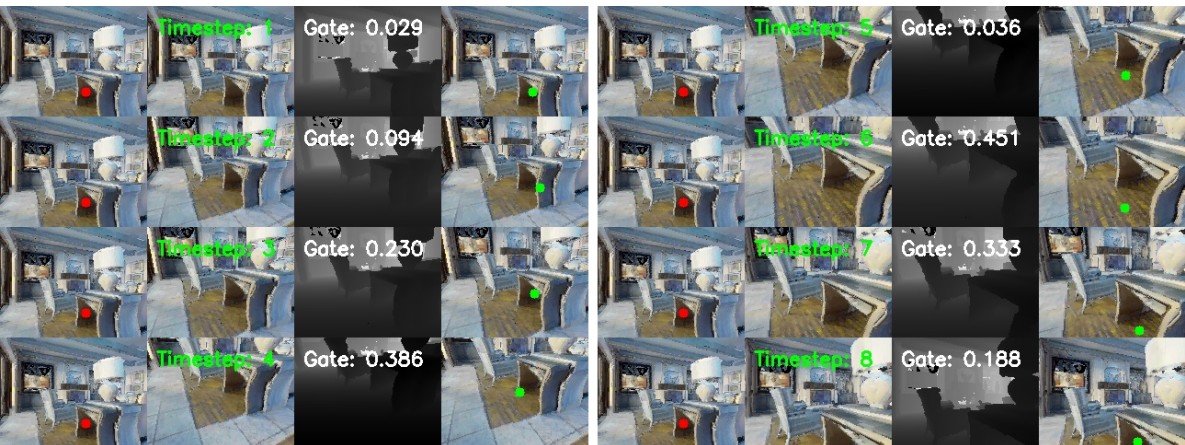

Figure 12: Analyzing the depth gate activations from an episode during rollout evaluation with HM3D with Causal SSM-RGBD PixNav with depth gate. The policy dynamically increases reliance on depth in geometrically complex or cluttered regions, while suppressing it when RGB observations are sufficient. Goal coordinate and predicted goal coordinate for an episode are marked as red and green circles respectively.

ambiguous regions (t=4–6), indicating increased reliance on depth. This demonstrates that the model adaptively modulates depth usage rather than relying on it uniformly.

