# OpenReview forum: "SSM-PixNav: State Space Models for Pixel-Guided Embodied Navigation"
_TMLR — Accepted by TMLR_

### Review · Reviewer_2Em8 · 2026-03-30

**Summary Of Contributions:**

The paper introduces SSM-PixNav, a framework for pixel-guided embodied navigation that utilizes State Space Models (SSMs) to improve upon existing transformer-based methods. The authors address three primary limitations in the field: the lack of geometric awareness in RGB-only policies, high computational costs of transformers, and the absence of open benchmark datasets.

Strengths:
1. Significant reduction in model parameters and inference latency (10-15ms faster) compared to transformers.
2. Improved robustness to Gaussian noise and varying history lengths.
3. Provision of an open-source dataset and code to address reproducibility issues.

**Additional Comments:**

This paper is novel and interesting. I would give a decision of "lean to accept" if my concerns can be addressed successfully.

**Audience:**

Yes

**Audience Explanation:**

TMLR's audience includes researchers in Embodied AI, Robotics, and Sequence Modeling. The paper’s application of Mamba to a practical robotics task navigation, provides valuable insights.

**Broader Impact Concerns:**

There are no significant ethical concerns. The paper uses standard photorealistic indoor datasets (HM3D, MP3D) for navigation. The authors have committed to releasing the code and dataset, which promotes transparency.

**Claims And Evidence:**

Yes

**Claims Explanation:**

The authors provide comprehensive empirical evidence through:

Comparative Analysis: Directly comparing SSM-based models against transformer baselines across multiple difficulty levels.

Ablation Studies: Testing the specific impact of the Mamba layer and varying history lengths

Cross-Dataset Validation: Demonstrating generalization by training on HM3D and testing on the MP3D dataset.

**Requested Changes:**

Clarification on Zero-Shot Failure: The RGBD-SSM models achieved 0% success in zero-shot object goal navigation (Hard). The authors should provide a more detailed discussion on why the introduction of depth data, which usually helps navigation, led to a total failure in this specific downstream task compared to RGB-only SSMs.

Consistency in Metric Reporting: Ensure all tables (specifically Table 3 vs Table 10) use consistent resolution or clearly explain the performance drop when moving from 120x160 to 224x224 where applicable.

Real-to-Sim Gap: While edge device latency is measured, adding a small qualitative section or plan for how the "sim-to-real gap" mentioned in the conclusion will be addressed would strengthen the practical claims.

Depth Gating Analysis: Provide a visualization or deeper data on the "learned mixing ratio" of the depth gate to show when the model chooses to rely on depth versus RGB.

---

> ### Author Response · Authors · 2026-04-27
> **Response to Reviewer 2Em8**
>
> We thank the reviewer for the valuable comments and constructive suggestions. The
>
> **1. Clarification on Zero-Shot Failure**
>
> We thank the author for this insightful concern, we tried to analyze it more closely. It was observed that the DINO predicted goal point for the agent to reach with PixNav is a different instance than the one pointed from the dataset. To this end if we see the distance to goal and distance to predicted goal are different. And a more significant point is that even though the iterations were not successful as per the dataset, our agent was successful in driving close to the predicted goal point (<1m). As the agent relies on external open source modules to detect the goal point, the success in reaching the predicted goal point can be indeed considered a success of the Pixel navigation agent. In other words, RGBD model allows the agent to reach another instance of goal in the object navigation setting, which can also be termed a success.
> <div align="center">
>
> | Model | Resolution | SR (↑) | SPL (↑) | SSPL (↑) | Distance To Goal (↓) | Distance To Predicted Goal (↓) |
> |:-----:|:----------:|:------:|:-------:|:--------:|:--------------------:|:------------------------------:|
> | PixNav* | 120×160 | 9 | 0.0517 | 0.0961 | 5.894 | 1.3238 |
> | PixNav (baseline) | 120×160 | 2 | 0.0200 | 0.0730 | 5.585 | 0.8667 |
> | SSM-RGB PixNav | 120×160 | **9** | **0.0638** | **0.1129** | **5.826** | 1.1713 |
> | Causal SSM-RGB PixNav | 120×160 | 3 | 0.0233 | 0.0937 | 5.560 | 1.0742 |
> | RGBD PixNav | 120×160 | 0 | 0 | 0.0011 | 5.627 | 0.8387 |
> | SSM-RGBD PixNav | 120×160 | 0 | 0 | 0.0079 | 5.538 | 0.8202 |
> | Causal SSM-RGBD PixNav | 120×160 | 0 | 0 | 0.0024 | 5.547 | 0.8202 |
> | Causal SSM-RGBD PixNav with depth gate | 120×160 | 0 | 0 | 0.0061 | 5.566 | 0.8205 |
>
> **Table:** **Zero-shot object goal navigation** results from Pix Nav models on 100 HM3D validation episodes.
>
> </div>
>
> **2. Consistency in Metric Reporting**
>
> We have changed  all metrics to have consistency in image resolution as 120x160. All models in our work are trained at a resolution of 120×160, following prior PixelNav setups. We observe a performance drop when evaluating at 224×224, which we attribute to a distribution mismatch between training and evaluation resolutions. In particular, learned visual representations, especially in models trained from scratch, are sensitive to input resolution, as changes in scale affect spatial feature statistics and receptive fields. Since our models are not fine-tuned at the higher resolution, this degradation is expected.
>
> **3. Real-to-Sim Gap**
>
>  Through our current experiments on edge devices we have only measured latency, but testing of the behaviour in real is pending where we need to address the sensor noise in data, control the bot as per the actions, etc. As suggested we have included an explanation of our future plan in the appendix  A.8
>
> **4.Depth Gating Analysis**
>
> To understand the depth gating more deeply we have plotted at each step the RGB observation and depth observation with depth ratio marked on it during rollout validation. A sample plot is shown below where goal and predicted coordinates are marked in red and green circles respectively. We have added an example as Figure 11 in revised manuscript for visual understanding of a sample trajectory.
> ![Image](https://drive.google.com/file/d/1shbWTQIcvivQYYJQ97TMPgPyoLuXTpXR/view?usp=drive_link)
> Here we can see the agent tried to leverage RGB only cues at the start t=1-2 but later we see the goal point being predicted inside the table/cluttered area. So the model starts to rely more on depth from t=3-5. Later we see the depth score slowly comes down as with RGB alone it will be able to handle the rest.

---

> > ### Comment · Action_Editor_4rFj · 2026-04-27
> >
> > Dear Reviewers,
> >
> > The authors have submitted their rebuttal. Please proceed with the discussion and finalize your recommendations. If any additional clarification is required, feel free to raise your questions to the authors.
> >
> > Thank you for your contributions to TMLR.
> >
> > Best,
> > AE

---

### Review · Reviewer_8haf · 2026-04-09

**Summary Of Contributions:**

Strengths
- Depth information is introduced into pixel-guided navigation, and an RGB-D version is proposed, where a gating mechanism is used to fuse RGB and depth features.
- A new open-source dataset, PixNav Trajectories, is constructed for pixel-guided navigation.

Weaknesses
- Incorporating depth information via channel concatenation is already a standard practice; this paper merely adopts a straightforward implementation, offering limited novelty.
- Mamba inherently models causality; why is it still necessary to explicitly introduce masking for causal modeling?
- The paper proposes a Depth Gate to aggregate RGB and depth information, but does not explain how the Depth Gate is computed or how it is learned.
- In the ablation study, Mamba is removed to demonstrate performance gains attributed to its capability in modeling dynamic information. However, why is there no comparison with other sequential models such as LSTM?
- Regarding inference speed, the baseline model has 54M parameters, while the SSM-based models have 27M and 38M parameters. Faster inference with fewer parameters is expected and may not be attributable to the architectural advantage itself.
- The dataset is constructed using SPF on the HM3D dataset for pixel-guided navigation, but there is a lack of evaluation regarding the dataset quality.
- In the robustness experiments, noise is only added to RGB images but not to depth. Without adding noise to depth, it is difficult to demonstrate that the RGB-D model is more robust than the RGB-only model.
- There is inconsistency between the textual description and the figures. In Section 3.2.1, the observation is described as a 384-dimensional vector after ResNet, convolution, and projection, whereas Figure 2 shows a 768-dimensional vector.
- Why does RGBD-PixNav use depth information when computing target features, while the SSM-RGBD PixNav Model and Causal SSM-RGBD PixNav do not utilize depth information?

**Audience:**

No

**Audience Explanation:**

The paper may be of limited interest to TMLR’s audience due to its relatively modest novelty, though some aspects could still be relevant to researchers working on embodied navigation.

**Claims And Evidence:**

Yes

**Claims Explanation:**

The claims are only partially supported, as some key design choices lack sufficient justification and several experimental evaluations are incomplete or unconvincing.

**Requested Changes:**

see weakness

---

> ### Author Response · Authors · 2026-04-27
> **Response to Reviewer 8haf**
>
> We thank the reviewer for the valuable comments and suggested key changes. The concerns posted are answered briefly below,
>
> **1. Incorporating depth information via channel concatenation is already a standard practice; this paper merely adopts a straightforward implementation, offering limited novelty.**
>
> We agree that channel-wise concatenation is a standard approach for incorporating additional modalities. Our intention was not to claim novelty in the fusion mechanism itself, but to investigate the role of depth in the pixel-level navigation setting, where prior work has predominantly focused on RGB-only inputs.
> To the best of our knowledge, the use of depth in this specific formulation of pixel-based navigation has not been systematically explored. We therefore begin with a simple and controlled integration via channel concatenation to isolate the effect of depth without introducing additional architectural complexity.
> Our contributions lie in (i) empirically demonstrating that depth provides consistent improvements in navigation performance and robustness, particularly in challenging scenarios, and (ii) showing that naive fusion is not sufficient, which motivates more structured designs. In line with this, our later models adopt a separate encoder and SSM-based temporal modeling to better exploit modality-specific features.
> We will revise the manuscript to clarify that the novelty is not in the concatenation itself, but in the analysis of depth as a modality in pixel navigation and the resulting design insights for more effective RGB-D integration.
>
> **2. Mamba inherently models causality; why is it still necessary to explicitly introduce masking for causal modeling?**
>
> While Mamba is architecturally causal, this property holds under sequential (autoregressive) execution. In our training setup, we process full trajectories (length 64) in parallel and predict actions for all timesteps jointly, meaning during training observations from all 64 steps are passed apriori. This introduces the possibility of future timestep information influencing earlier predictions through parallel computation and gradient flow. Therefore, we introduce explicit causal masking to enforce that predictions at time t depend only on inputs up to t, aligning training with the intended autoregressive formulation.
>
> **3. The paper proposes a Depth Gate to aggregate RGB and depth information, but does not explain how the Depth Gate is computed or how it is learned.**
> We regret the lack of clarity on how the depth gate decides the ratio of depth features to be passed to the model. In Causal-RGBD SSM PixNav, we see the performance is not close to the RGB variant but it is better than other RGBD variants. As depth may not always be useful in making the decision and we wanted the RGB to dominate, so the gate is introduced to learn the fraction of depth token being passed to the Mamba model.
>
>        gate = self.depth_gate(torch.cat([rgb_tok, dep_tok], dim=-1))
>        gate = 0.5 * gate  dep_tok = gate * dep_tok
> We employ a gated fusion mechanism where depth features are modulated by a learned gate conditioned on both RGB and depth tokens. The gate is explicitly scaled to limit the maximum contribution of depth, ensuring stable training and preventing modality dominance. We have added this in the paper too.
> **4. In the ablation study, Mamba is removed to demonstrate performance gains attributed to its capability in modeling dynamic information. However, why is there no comparison with other sequential models such as LSTM?**
> We agree that comparison with other sequential models such as LSTMs is valuable. Our primary comparisons focused on Transformer-based baselines, as they represent the dominant paradigm in prior PixNav work. We chose Mamba due to its ability to model long-range dependencies with linear complexity. We have included LSTM-based comparisons in the revised version appendix A.5 to provide a more comprehensive evaluation of sequential modeling choices.
> <div align="center">
>
> | Model | Dataset | Easy SR (↑) | Easy SPL (↑) | Medium SR (↑) | Medium SPL (↑) | Hard SR (↑) | Hard SPL (↑) |
> |:---:|:---:|:---:|:---:|:---:|:---:|:---:|:---:|
> | **Causal SSM-RGB PixNav** | HM3D | **0.8043** | **0.7905** | **0.4808** | **0.4727** | **0.2273** | **0.2273** |
> | **Causal SSM-RGB PixNav** | MP3D | 0.6418 | 0.6277 | **0.2128** | **0.2090** | **0.0968** | **0.0968** |
> | Causal LSTM-RGB PixNav | HM3D | 0.7764 | 0.7685 | 0.2826 | 0.2781 | 0.1500 | 0.1451 |
> | Causal LSTM-RGB PixNav | MP3D | **0.6598** | **0.6484** | 0.1700 | 0.1696 | 0.0800 | 0.0800 |
>
> **Table:** Pixel navigation policy evaluation results in the HM3D datasets. The model used here is the same as the proposed Causal SSM-RGB PixNav, but Mamba layers are skipped.
>
> </div>

---

> > ### Author Response · Authors · 2026-04-27
> > **Response to Reviewer 8haf**
> >
> > **4. Regarding inference speed, the baseline model has 54M parameters, while the SSM-based models have 27M and 38M parameters. Faster inference with fewer parameters is expected and may not be attributable to the architectural advantage itself.**
> >
> > We agree that reduced parameter count contributes to improved inference speed. Our key point is that the proposed Mamba-based models simultaneously achieve improved performance and reduced model size compared to the baseline. This is particularly relevant in embodied settings, where multiple modules run concurrently on resource-constrained hardware. To reflect this, we report inference time on edge devices, highlighting the practical efficiency of our approach beyond parameter count alone.
> >
> > **5. The dataset is constructed using SPF on the HM3D dataset for pixel-guided navigation, but there is a lack of evaluation regarding the dataset quality.**
> >
> > We thank the reviewer for pointing this out. While our initial submission did not explicitly report dataset quality metrics, the dataset is generated using a Shortest Path Follower (SPF) oracle on the HM3D dataset, which is a standard and widely adopted approach in prior works listed in references. [1] explicitly introduces Oracle success rate as a metric.
> > To quantitatively assess dataset quality, we evaluate the oracle policy itself under the PixNav rollout setting. The oracle agent achieves success rates of 91.87% (easy), 93.10% (medium), and 57.14% (hard). These results indicate that the generated trajectories are largely feasible and provide a strong supervisory signal, while the lower performance on hard episodes reflects increased task complexity rather than noise in the data.
> >
> > Additionally, since trajectories are derived from geodesically optimal paths, the dataset ensures:
> > (i) high-quality action supervision,
> > (ii) consistent goal-directed behavior, and
> > (iii) coverage across varying difficulty levels in realistic indoor environments.
> > We will revise the manuscript to explicitly include these evaluations and clarify the dataset construction protocol and its reliability.
> >
> > References:
> > 1. Wang, X., Huang, Q., Celikyilmaz, A., Gao, J., Shen, D., Wang, Y.F., Wang, W.Y. and Zhang, L., 2019. Reinforced cross-modal matching and self-supervised imitation learning for vision-language navigation. In Proceedings of the IEEE/CVF conference on computer vision and pattern recognition (pp. 6629-6638).
> > 2. Cai, W., Huang, S., Cheng, G., Long, Y., Gao, P., Sun, C. and Dong, H., 2024, May. Bridging zero-shot object navigation and foundation models through pixel-guided navigation skill. In 2024 IEEE International Conference on Robotics and Automation (ICRA) (pp. 5228-5234). IEEE.
> > 3. Krishnan R, A. and Channappayya, S.S., 2025, December. TEPEN: Towards an Ensemble Model for Pixel-Based Embodied Navigation. In International Conference on Pattern Recognition and Machine Intelligence (pp. 60-67). Cham: Springer Nature Switzerland.
> >
> > **6. There is inconsistency between the textual description and the figures. In Section 3.2.1, the observation is described as a 384-dimensional vector after ResNet, convolution, and projection, whereas Figure 2 shows a 768-dimensional vector.**
> >
> > We thank the reviewer for pointing this out. Each modality is projected from 512D to 384D, and the concatenated representation results in a 768D feature vector. Figure 2 contained an inconsistency, which has been corrected in the revised version.

---

> ### Author Response · Authors · 2026-04-27
> **Response to Reviewer 8haf**
>
> **7. In the robustness experiments, noise is only added to RGB images but not to depth. Without adding noise to depth, it is difficult to demonstrate that the RGB-D model is more robust than the RGB-only model.
> We thank the reviewer for this observation. In the current version, noise was applied only to RGB inputs. We agree that this does not fully evaluate robustness in the RGB-D setting. In the revised version, we have extended the experiments to include noise perturbations in depth maps as well, providing a more comprehensive robustness analysis. We see that the performance degrades as depth is also mixed with noise. The performance degradation with the increase in noise level is mainly due the distortion in the structural cues Table is updated as Table 5 in revised manuscript.
> <div align="center">
>
> | Model | Std. Dev. | Easy SR (↑) | Easy SPL (↑) | Medium SR (↑) | Medium SPL (↑) | Hard SR (↑) | Hard SPL (↑) |
> |:---:|:---:|:---:|:---:|:---:|:---:|:---:|:---:|
> | PixNav* (Author's weights) | 0.02 | 0.1005 | 0.0853 | 0.0000 | 0.0000 | 0.0500 | 0.0500 |
> || 0.03 | 0.0916 | 0.0774 | 0.0192 | 0.0163 | 0.1364 | 0.1364 |
> || 0.04 | 0.1038 | 0.0839 | 0.0189 | 0.0189 | 0.0800 | 0.0800 |
> || 0.05 | 0.1110 | 0.0945 | 0.0000 | 0.0000 | 0.0769 | 0.0769 |
> | RGB-PixNav (baseline) | 0.02 | 0.3643 | 0.3398 | 0.1500 | 0.1478 | 0.1176 | 0.1176 |
> || 0.03 | 0.3501 | 0.3288 | 0.0833 | 0.0833 | 0.2400 | 0.2400 |
> || 0.04 | 0.3470 | 0.3273 | 0.1250 | 0.1224 | 0.1579 | 0.1579 |
> || 0.05 | 0.3530 | 0.3326 | 0.0784 | 0.0784 | 0.2500 | 0.2500 |
> | SSM-RGB PixNav | 0.02 | 0.7459 | 0.7350 | 0.4681 | 0.4661 | 0.1739 | 0.1739 |
> || 0.03 | 0.7076 | 0.6982 | 0.2500 | 0.2481 | 0.0000 | 0.0000 |
> || 0.04 | 0.6843 | 0.6750 | 0.2807 | 0.2803 | 0.0000 | 0.0000 |
> || 0.05 | 0.6507 | 0.6391 | 0.3000 | 0.3000 | 0.0870 | 0.0870 |
> | Causal SSM-RGB PixNav | 0.02 | **0.7596** | **0.7466** | **0.3600** | **0.3505** | **0.1852** | **0.1852** |
> || 0.03 | 0.7595 | 0.7442 | 0.2727 | 0.2671 | 0.0952 | 0.0952 |
> || 0.04 | 0.7299 | 0.7203 | 0.2807 | 0.2771 | 0.1429 | 0.1429 |
> || 0.05 | 0.7051 | 0.6909 | 0.2679 | 0.2624 | 0.1562 | 0.1562 |
> | RGBD-PixNav | 0.02 | 0.2674 | 0.2297 | 0.1026 | 0.1026 | 0.0000 | 0.0000 |
> || 0.03 | 0.3084 | 0.2536 | 0.0357 | 0.0357 | 0.0000 | 0.0000 |
> || 0.04 | 0.2331 | 0.1941 | 0.0000 | 0.0000 | 0.0000 | 0.0000 |
> || 0.05 | 0.2072 | 0.1804 | 0.0851 | 0.0851 | 0.0000 | 0.0000 |
> | SSM-RGBD PixNav | 0.02 | 0.4213 | 0.3838 | 0.1702 | 0.1629 | 0.1200 | 0.1200 |
> || 0.03 | 0.3939 | 0.3624 | 0.1509 | 0.1509 | 0.0455 | 0.0455 |
> || 0.04 | 0.4262 | 0.3863 | 0.1667 | 0.1572 | 0.0833 | 0.0833 |
> || 0.05 | 0.4246 | 0.3874 | 0.1702 | 0.1682 | 0.0400 | 0.0400 |
> | Causal SSM-RGBD PixNav | 0.02 | 0.6750 | 0.6519 | 0.1875 | 0.1806 | **0.1538** | **0.1538** |
> || 0.03 | 0.6446 | 0.6254 | 0.2830 | 0.2762 | 0.2222 | 0.2222 |
> || 0.04 | 0.6317 | 0.6129 | 0.2364 | 0.2348 | 0.2609 | 0.2609 |
> || 0.05 | 0.6178 | 0.5988 | 0.1136 | 0.1136 | 0.1739 | 0.1739 |
> | Causal SSM-RGBD PixNav with Depth gate | 0.02 | **0.7092** | **0.6860** | **0.2500** | **0.2413** | 0.0690 | 0.0690 |
> || 0.03 | 0.6494 | 0.6323 | 0.2444 | 0.2356 | 0.1579 | 0.1579 |
> || 0.04 | 0.6093 | 0.5901 | 0.3061 | 0.2996 | 0.0625 | 0.0625 |
> || 0.05 | 0.6071 | 0.5888 | 0.3191 | 0.3148 | 0.1429 | 0.1429 |
> **Table:** Robustness of proposed PixNav policies towards Gaussian Noise of mean zero.
> </div>

---

> > ### Author Response · Authors · 2026-04-27
> > **Response to Reviewer 8haf**
> >
> > **8.Why does RGBD-PixNav use depth information when computing target features, while the SSM-RGBD PixNav Model and Causal SSM-RGBD PixNav do not utilize depth information?**
> > We agree that incorporating depth in the goal representation benefits Transformer-based RGBD-PixNav models. While Transformers can encode temporal context in observation tokens, they still rely on cross-attention for direct goal–observation matching, which benefits from geometry (depth) being present in both. In contrast, SSMs compress temporal information into a latent state that accumulates geometry from observations.
> > As a result, RGB goal descriptors are sufficient, and depth is only required in observation encoding.In our SSM-based models, RGB goal descriptors alone are sufficient for effective goal localization, while depth information in the observation stream provides the necessary geometric cues for navigation. This is consistent with the sequential nature of SSMs, which can accumulate spatial structure over time from observations rather than relying on explicit geometric information in the goal.
> > Based on this, we restrict depth usage to observation encoding, which slightly simplifies the model while still achieving strong performance compared to RGBD Transformer baselines.
> > As an ablation we have tried using RGBD goal descriptors as used in transformer based models on Causal SSM-RGBD PixNav with depth gate, the metrics confirms our assumptions above. Depth in goal descriptor is not aiding especially in SSM based variants.
> > <div align="center">
> >
> > | Model | Dataset | Easy SR (↑) | Easy SPL (↑) | Medium SR (↑) | Medium SPL (↑) | Hard SR (↑) | Hard SPL (↑) |
> > |:---:|:---:|:---:|:---:|:---:|:---:|:---:|:---:|
> > | Causal SSM-RGB PixNav with RGBD goal descriptor | HM3D | 0.4242 | 0.4161 | 0.0545 | 0.0545 | 0.1111 | 0.1111 |
> > |  | MP3D | 0.4902 | 0.4798 | 0.0306 | 0.0306 | 0.0606 | 0.0606 |
> >
> > **Table:** Performance of Goal depth aware variant of Causal SSM RGBD PixNav with depth gate during policy rollout.
> >
> > </div>

---

> ### Comment · Action_Editor_4rFj · 2026-04-27
>
> Dear Reviewers,
>
> The authors have submitted their rebuttal. Please proceed with the discussion and finalize your recommendations. If any additional clarification is required, feel free to raise your questions to the authors.
>
> Thank you for your contributions to TMLR.
>
> Best,
> AE

---

> > ### Comment · Action_Editor_4rFj · 2026-05-09
> >
> > Dear Reviewer,
> >
> > The authors have submitted their rebuttal. Please proceed with the discussion and finalize your recommendations. If any additional clarification is required, feel free to raise your questions to the authors.
> >
> > Thank you for your contributions to TMLR.
> >
> > Best, AE

---

### Review · Reviewer_Y13a · 2026-04-20

**Summary Of Contributions:**

The paper “SSM-PixNav: State Space Models for Pixel-Guided Embodied Navigation” proposes a new approach to robot navigation that uses pixel-level goal guidance combined with efficient sequence modeling to improve accuracy and scalability. Instead of relying only on RGB images or heavy transformer architectures, the authors introduce models that incorporate depth information (RGBD) and leverage lightweight State Space Models (specifically Mamba) to capture temporal dependencies more efficiently. They also create a new public dataset (PixNav Trajectories) to address reproducibility gaps in prior work. Through extensive experiments in simulated environments, the proposed causal SSM-based models significantly outperform transformer baselines—achieving higher navigation success rates, better robustness to noise, and reduced model size—while maintaining real-time efficiency, making them well-suited for deployment on resource-constrained robotic systems.

The works lays on Cai et al 2023. Methodologically, the authors retain key elements from Cai et al.—such as the multi-head objective (action, distance, and tracking losses), the use of goal masks, and supervised trajectory learning—but extend them by (1) incorporating depth as a first-class input rather than a secondary cue, (2) replacing transformer-based temporal modeling with more efficient state-space models (Mamba), and (3) introducing a new benchmark dataset to standardize evaluation. Overall, the work can be seen as both a reproduction study and a systematic improvement over Cai et al., exposing weaknesses in the original pipeline (data opacity, computational cost, limited geometric awareness) while preserving its core pixel-navigation paradigm. The current paper explicitly attempts to reproduce Cai et al.’s setup, but reports that the original pretrained weights perform poorly on the newly constructed dataset, requiring re-training and leading to significantly different results, suggesting dataset dependence and limited generalization.


Strengths:

1.	The authors provide improvement over existing SoTA

2.	Significant empirical validation with ablation is the right way to go on TMLR!

3.	Contributes an important data set to validate results in the area

Weaknesses:

1.	The paper can benefit from a revision so that instead of extensive technical description of what is in the images could bring to the front the advantages of the components that are novel in it vs Cai et al, and underline their importance in the overall process.
2.	The unfamiliar reader will be thrown into a problem that is not defined in the body of the paper clearly. Unless one reads Cai et a. its is hard to understand the true story behind the construction the authors propose. Please dedicate a subsection for the problem definition. For example, it was not clear to me that the target pixels are necessarily available on every timestep. Also, an example problem, as Cai et al suggest, would do good.
3.	Some level of formulation would do good as well – the concept of causality is not clear in the intro. Only later we understand that it relates to the attention. Its hard to understand what are ‘the three heads’ in the related work
4.	The text font size in the images is unreadable.
5.	It is not clear what the Max trajectory length leading to causal attention and position embedding at the bottom of Fig 1.
6.	In Figure 4: if we didn’t reach the maximum timesteps, we are generating a new pixel goal? What sense does it make? For episode it does makes sense, but for every step? why?
7.	Results in Table3: how come better results are achieved without incorporating depth in easy and medium cases?
8.	Results in Table 4: how come depth gate doesn’t help achive good results, except partially in the Easy case?
9.	In section 5.3 is the addition of noise aimed to the pixels RGBD values? If so, why did you choose such small factors of the STD? what happens when you add an order of magnitude more: 0.2, for example…
10.	Table 5: the values of Causal RBBG with gate are not better than the other values for the Hard problem – why did you highlight them? 0.0500 is the worse result!
11.	Discussion – it is not clear what does the 17% relate to.
12.	Conclusion - no mention of the mamba speedup?

**Audience:**

Yes

**Audience Explanation:**

The problem tackled inthis paper is of high inportance to emobided AI research, computational vision, and more

**Broader Impact Concerns:**

no concerns

**Claims And Evidence:**

Yes

**Claims Explanation:**

Empirical evaluation is sufficient, includes ablation studies

**Requested Changes:**

please see weaknesses above

---

> ### Author Response · Authors · 2026-04-27
> **Response to Reviewer Y13a**
>
> We thank the reviewer for the valuable comments and suggested key changes. The concerns posted are answered briefly below,
>
> **1. The paper can benefit from a revision so that instead of extensive technical description of what is in the images could bring to the front the advantages of the components that are novel in it vs Cai et al, and underline their importance in the overall process.**
>
> We thank the reviewer for this feedback. We have only explained the RGB pixNav in more detail to set the context. All the subsequent sections present the incremental steps we have introduced to the existing framework. The figure caption has been adjusted in the revised manuscript.
>
> **2. The unfamiliar reader will be thrown into a problem that is not defined in the body of the paper clearly. Unless one reads Cai et a. its is hard to understand the true story behind the construction the authors propose. Please dedicate a subsection for the problem definition. For example, it was not clear to me that the target pixels are necessarily available on every timestep. Also, an example problem, as Cai et al suggest, would do good.**
>
> Again, this is a valuable comment about providing the big picture. We have revised the Proposed Models for Pixel-Guided Navigation section to define the problem statement upfront before explaining each approach.
>
> **3. Some level of formulation would do good as well – the concept of causality is not clear in the intro. Only later we understand that it relates to the attention. Its hard to understand what are ‘the three heads’ in the related work**
>
> We have introduced the need for causality in Appendix. As suggested, we have now provided more details on the 3 heads in Related works too.
>
> **4. The text font size in the images is unreadable.**
>
> We have increased the font size especially in Fig 1 as it was less legible as rightly pointed out. We thank the reviewer for pointing this out.
>
> **5. It is not clear what the Max trajectory length leading to causal attention and position embedding at the bottom of Fig 1.**
>
> Again, thanks for pointing to this important detail. Max trajectory length is for the number of steps taken. For each step taken for the transformer PE is created with step information gives the order of the tokens. Causal Attention and target mask is for mask generation to ensure causality. Thanks for pointing it out, causality is part of target mask generation. The figure is updated.
>
> **6. In Figure 4: if we didn’t reach the maximum timesteps, we are generating a new pixel goal? What sense does it make? For episode it does makes sense, but for every step? why?**
>
> We regret the confusion. We have updated the figure in the revised manuscript. Goal is sampled once every episode.
>
> **7. Results in Table3: how come better results are achieved without incorporating depth in easy and medium cases?**
>
> We thank the reviewer for this observation. We note that in easy and medium scenarios, RGB-only models already achieve strong performance because the goal is often visually salient and reachable with short-horizon reasoning. In such cases, appearance cues alone are sufficient for accurate action prediction, and the additional geometric information from depth provides limited marginal benefit.
> Moreover, depth signals can be noisy and less informative in near-field or low-variation regions (e.g., flat surfaces or clutter), which may introduce ambiguity if not carefully integrated. This is reflected in our experiments, where naive fusion does not consistently improve performance.
> However, in harder scenarios, where spatial reasoning and long-range structure become more critical, incorporating depth leads to improved performance. Our gated fusion design further supports this observation by allowing the model to selectively utilize depth information rather than relying on it uniformly.
> We will clarify this behavior and provide additional discussion in the revised manuscript.

---

> > ### Author Response · Authors · 2026-04-27
> > **Response to Reviewer Y13a**
> >
> > **8. Results in Table 4: how come depth gate doesn’t help achieve good results, except partially in the Easy case?**
> >
> > We thank the reviewer for this observation. In the cross-dataset setting (HM3D → MP3D), the effectiveness of depth-based fusion is influenced by domain shift between the training and evaluation environments.
> > While the depth gating mechanism improves performance on HM3D, it does not consistently translate to MP3D. This is likely because depth distributions, scene layouts, and sensor characteristics differ across the two datasets. As a result, the learned gating parameters, which adaptively weight depth based on HM3D statistics, may not generalize well to MP3D.
> > In contrast, RGB features tend to be more invariant to such domain shifts, allowing RGB-only or RGB-dominant models to generalize better in some cases. The partial improvement observed in the Easy setting can be attributed to simpler scene structure, where depth cues remain relatively consistent across datasets.
> > We will clarify this limitation and highlight the impact of cross-dataset distribution shift on depth utilization in the revised manuscript.
> >
> > **9. In section 5.3 is the addition of noise aimed to the pixels RGBD values? If so, why did you choose such small factors of the STD? what happens when you add an order of magnitude more: 0.2, for example…**
> >
> > We have increased noise from 2-5%, we have introduced noise in the RGB and depth . If we further increase the noise it affects the perceived geometry and features which will degrade the performance as expected.
> >
> > **10. Table 5: the values of Causal RBBG with gate are not better than the other values for the Hard problem – why did you highlight them? 0.0500 is the worse result!**
> >
> > Even now, in the hard case, Causal SSM-RGBD PixNav with depth gate is not the best. So we have highlighted for hard the best Causal SSM-RGBD PixNav.
> >
> > **11.Discussion – it is not clear what does the 17% relate to.**
> >
> > It relates to the inference time speed up on the edge compute. This has been clarified in the revised manuscript.
> >
> > **12. Conclusion - no mention of the mamba speedup?**
> >
> > Thanks for catching this miss. We have added this speedup to the Conclusion section.

---

> ### Comment · Action_Editor_4rFj · 2026-04-27
>
> Dear Reviewers,
>
> The authors have submitted their rebuttal. Please proceed with the discussion and finalize your recommendations. If any additional clarification is required, feel free to raise your questions to the authors.
>
> Thank you for your contributions to TMLR.
>
> Best,
> AE

---

> > ### Comment · Action_Editor_4rFj · 2026-05-09
> >
> > Dear Reviewer,
> >
> > The authors have submitted their rebuttal. Please proceed with the discussion and finalize your recommendations. If any additional clarification is required, feel free to raise your questions to the authors.
> >
> > Thank you for your contributions to TMLR.
> >
> > Best, AE

---

> > > ### Comment · Reviewer_Y13a · 2026-05-09
> > > **Lean to accept**
> > >
> > > Im happy with the way the authors addressed my comments in the revision, and I recommedn accepting the paper

---

### Review · Reviewer_SCQw · 2026-04-20

**Summary Of Contributions:**

## Summary
This paper introduces **SSM-PixNav**, a framework for pixel-guided embodied navigation that addresses limitations of prior RGB-only, transformer-based approaches by incorporating depth information and leveraging **state space models (SSMs), specifically Mamba**, for efficient temporal modeling. The authors propose a suite of models, including RGB, RGBD, and causal SSM-based variants with a depth-gating mechanism, trained via imitation learning on a newly curated **PixNav Trajectories dataset** derived from HM3D in Habitat-sim. The method formulates navigation as a multi-head prediction problem (action, distance, and pixel tracking) and replaces quadratic-cost attention with linear-time SSMs to improve efficiency. Empirical evaluations demonstrate that **causal SSM-based policies outperform transformer baselines**, achieving higher success rates and SPL while reducing model size by roughly half, and exhibiting improved robustness to observation noise and varying temporal context. Additionally, the work emphasizes reproducibility through dataset release and shows moderate cross-dataset generalization to MP3D environments.
## Strengths
* The paper tackles a well-motivated problem and clearly identifies limitations in prior pixel navigation work (RGB-only inputs, transformer inefficiency, lack of reproducible datasets).
* Introducing Mamba-based state space models for navigation is a meaningful and technically sound contribution, especially given the sequential nature of the task.
* The use of depth (RGBD) is well-justified, and the depth-gating mechanism is a thoughtful addition beyond simple feature fusion.
* The causal formulation aligns well with navigation and is implemented consistently across models.
* The empirical results show strong improvements over baselines, both in performance (SR/SPL) and efficiency (smaller models, faster inference).
* The experiments are fairly comprehensive, including ablations, robustness to noise, and sensitivity to history length.

## Weakness
* The main novelty is somewhat incremental, as it largely combines existing components (RGBD inputs, imitation learning, Mamba/SSMs) rather than introducing a fundamentally new formulation of the problem.
* The training setup relies purely on supervised imitation learning from an oracle, which limits realism and raises concerns about compounding errors and deployment in interactive settings.
* The dataset, while useful, is generated from a shortest-path follower, which may bias the policies toward ideal trajectories and reduce diversity in behaviors.
* Comparisons to prior work are not entirely convincing and limited.
* The claimed generalization is relatively weak, as cross-dataset evaluation is limited and still within similar indoor simulation environments.
* The depth-gating mechanism is not deeply analyzed, and it is unclear when and why it helps or hurts performance.
* The paper lacks comparison to stronger or more recent baselines (e.g., RL-based or VLM-based navigation methods), which limits the strength of the empirical claims.
* The zero-shot navigation results are quite weak, suggesting limited applicability beyond the supervised setting.
* Some implementation and training details are not fully discussed, which may still hinder full reproducibility despite the dataset contribution.

**Audience:**

Yes

**Audience Explanation:**

They paper is still insightful and can help the audience to learn from the proposed exploration of the domain. Though it needs major restructuring and more revision to enhance the narration flow of the idea.

**Broader Impact Concerns:**

There are no "Broader Impact Concerns" for this paper.

**Claims And Evidence:**

No

**Claims Explanation:**

The paper needs to address the weaknesses listed above and the requested changes listed below to provide supporting evidence towards the claims.

**Requested Changes:**

- The methodology section needs major revision in which problem formulation, background, and then novelty are strongly intermixed and it is hard for the audience to delineate the novelties from the backgrounds.
- It is not clear in the transformer and SSM encoder model, what the novelty of this paper is? It seems the RGB+D encoding network is very common and trivial.
- The outputs heads needs to be elaborated and the final objective (loss) function is not defined and discussed.
- Comparison with VLM/VLA and LLM-based policies are not discussed. Overall, the experimental evaluation is very narrow and limited. The question is why don't we have the SoTA methods compared with the proposal on the embodied task? is the task different from those in the embodied benchamrk evaluation datasets?
- Can you elaborate on if the embodied task is pixel navigation, image, navigation, or object navigation?
- We need clarity on how the training and validation sets are created and partitioned?
- The paper’s novelty claim around RGBD input should be narrowed and made precise. Can you revisit that the claim of this paper is to study end-to-end RGBD fusion within PixNav-style transformer/SSM policies, rather than being the first to consider depth in the broader pixel-navigation literature.
- Can you elaborate why the performance of the "Causal SSM-RGB PixNav" is higher than "Causal SSM-RGBD PixNav" in Table 3?
- The details on the network and optimization hyper-parameters for re-generation purposes are missing.

---

> ### Author Response · Authors · 2026-04-27
> **Response to Reviewer SCQw**
>
> We thank the reviewer for the valuable comments and suggested key changes. The concerns posted are answered briefly below,
>
> **1. The methodology section needs major revision in which problem formulation, background, and then novelty are strongly intermixed and it is hard for the audience to delineate the novelties from the backgrounds.**
>
> We thank the reviewer for this feedback. While the contributions are summarized in the introduction, we agree that the methodology section can be structured more clearly. We have updated the methodology part in the revised draft.
>
> **2. It is not clear in the transformer and SSM encoder model, what the novelty of this paper is? It seems the RGB+D encoding network is very common and trivial.**
>
> Your observation is correct to encode all observations we have used Resnet 18 encoder. This was observed as the better with initial ablations with the prior work. For transformer models respecting the steps taken by prior works, we tried extending their work to reproduce the results and use those weights to aid the RGBD training. In SSM based policies we have used pretrained weights of encoders which indeed aided the success rate to improve, almost 2 times the transformer based baseline.
> The outputs heads needs to be elaborated and the final objective (loss) function is not defined and discussed.
> The outputs are discussed in the RGB PixNav model. We kindly refer the reviewer to Section 3.1. We avoided repeating the same lines as the heads are same and their learning is ensured through defining the loss terms. The final objective is discussed in the revision.
>
>  **3. The outputs heads needs to be elaborated and the final objective (loss) function is not defined and discussed.**
>
> The outputs are discussed in the RGB PixNav model. We kindly refer the reviewer to Section 3.1. We avoided repeating the same lines as the heads are same and their learning is ensured through defining the loss terms. The final objective is discussed in the revision.
>
> **4. Comparison with VLM/VLA and LLM-based policies are not discussed. Overall, the experimental evaluation is very narrow and limited. The question is why don't we have the SoTA methods compared with the proposal on the embodied task? is the task different from those in the embodied benchamrk evaluation datasets?**
>
> In PixNav the SOTA method is the one we have discussed as the baseline. The research here was limited due to availability of dataset to train. Through this work we are releasing the dataset to advance the research towards this problem too. Cited works of the prior work leveraged mostly the use of VLM’s in decision making not Pixel navigation method.  If we compare them the evaluation may not be fair. In Appendix we took similar task of goal instance navigation SOTA work and compared it against our performance.
>
> **5.Can you elaborate on if the embodied task is pixel navigation, image, navigation, or object navigation?**
>
> Embodied navigation as rightly mentioned can be of different types as image goal navigation, object goal navigation,pixel goal navigation. Through this paper we propose models for pixel goal navigation where the agent is asked to go to a image goal which is defined more  precisely by a binary mask representing the goal coordinates. We also compare the performance of the trained  model towards zero shot object goal navigation. We also compare our results with the  sota of a related problem statement, goal instance navigation.
>
> **6.We need clarity on how the training and validation sets are created and partitioned?**
>
> Training set and validation set generation pipeline is included where we leverage SPF agent to curate the ground truth. The balance of different complexity levels is ensured. Partition is ensured by taking the samples from train and eval folds of HM3D scenes. The goal points are randomly sampled, and are not reused from HM3D scene definition.
>
> **7. The paper’s novelty claim around RGBD input should be narrowed and made precise. Can you revisit that the claim of this paper is to study end-to-end RGBD fusion within PixNav-style transformer/SSM policies, rather than being the first to consider depth in the broader pixel-navigation literature.**
> To the best of our knowledge in pixel navigation task we are first to introduce RGBD at model level. Earlier people have attempted RGBD but to other types of embodied navigation like point goal navigation, object goal navigation and image goal navigation.

---

> > ### Author Response · Authors · 2026-04-27
> > **Response to Reviewer SCQw**
> >
> > **8. Can you elaborate why the performance of the "Causal SSM-RGB PixNav" is higher than "Causal SSM-RGBD PixNav" in Table 3?**
> >
> > If we see RGB models are performing better than the RGBD variants in SSM variants. But these models are better than the transformer variant. So depth is geometry information, as there is no guidance when to look at the depth  in our approach depth would have confused the model when fused in full. We see by introducing a depth gate which learns how much depth data to be learned, the performance is further improved.
> >
> > **9.The details on the network and optimization hyper-parameters for re-generation purposes are missing.**
> >
> > For all the trained variants, we refer the reviewer to Appendix A.1 where we have presented these details.

---

> > > ### Comment · Action_Editor_4rFj · 2026-04-27
> > >
> > > Dear Reviewers,
> > >
> > > The authors have submitted their rebuttal. Please proceed with the discussion and finalize your recommendations. If any additional clarification is required, feel free to raise your questions to the authors.
> > >
> > > Thank you for your contributions to TMLR.
> > >
> > > Best,
> > > AE

---

> > > > ### Comment · Action_Editor_4rFj · 2026-05-09
> > > >
> > > > Dear Reviewer,
> > > >
> > > > The authors have submitted their rebuttal. Please proceed with the discussion and finalize your recommendations. If any additional clarification is required, feel free to raise your questions to the authors.
> > > >
> > > > Thank you for your contributions to TMLR.
> > > >
> > > > Best, AE

---

### Author Response · Authors · 2026-04-27
**Summary of changes in the revised manuscript**

We sincerely thank the reviewers for their constructive feedback and insightful suggestions. We have carefully revised the manuscript to address all concerns and improve clarity, rigor, and completeness.

First, we have restructured the methodology section to clearly separate problem formulation, background, and our proposed contributions. The revised version improves readability and makes the novelty of our approach more explicit.

To address concerns regarding dataset quality, we now include an explicit evaluation of the SPF-based oracle used for dataset generation. Following prior work, we report oracle success rates across difficulty levels, demonstrating that the generated trajectories provide strong and reliable supervision signals.

For the robustness analysis, we agree that evaluating only RGB perturbations was insufficient. We have extended our experiments to include noise in both RGB and depth modalities, providing a more comprehensive assessment. The results show consistent degradation with increasing noise, highlighting the sensitivity of depth signals and motivating our design choices.

Regarding depth integration, we acknowledge that channel concatenation is a standard approach. Our contribution lies not in proposing a new fusion mechanism, but in systematically evaluating depth in the pixel-navigation setting, where it has been largely unexplored. Furthermore, we extend this baseline with SSM-based architectures and depth gating, offering deeper insights into when and how depth is beneficial.

An LSTM variant was trained and evaluated as per the reviewer comment, as expected this ablation have shown that LSTM help in short horizon or easier navigation task.

Our plan on future work -sim to real covers testing the behaviour on a physical robot addressing the related challenges. This is explained in the revised draft.

To understand the need for Causality, effect of depth gate separate subsections are introduced in Appendix to help the reader understand the concept better.

Captioning, Figure related concerns are addressed.

We have also clarified observations from our experiments:

1. RGB-only models perform strongly in simpler settings, indicating that appearance cues are often sufficient.

2. Depth can introduce noise or redundancy if not handled carefully, particularly in easier scenarios.

3. Our gating-based formulation helps regulate this effect, demonstrating that selective use of depth is more effective than naïve fusion.

Finally, we have improved the presentation of results, corrected inconsistencies, and clarified ablations to ensure that all claims are well-supported. We thank each reviewer as we believe that the each review comment was valuable and helped us improve our paper quality.

---

### Author Response · Authors · 2026-05-01
**Gentle reminder : Revisions and Responses**

Dear AE and Reviewers,

We have submitted our detailed responses addressing all four reviewers' comments. As the discussion period is nearing its end, we would be grateful if you could review the updates at your convenience.

---

> ### Comment · Action_Editor_4rFj · 2026-05-02
>
> I have sent reminders to all the reviewers. Please wait for their feedbacks. Thanks, AE

---

### Decision · Action_Editor_4rFj · 2026-05-23

**Recommendation:** Accept with minor revision

**Additional Comments:**

We received four reviews, with one reviewer (Reviewer SCQw) leaning toward rejection due to the current presentation quality of the paper. The comments from the other reviewers have been fully addressed. Given the overall contribution and significance of the work, the Action Editor recommends acceptance pending minor revisions.

The revision should specifically incorporate Reviewer SCQw’s feedback and substantially improve the presentation of the paper. In particular, while the idea is novel and relevant to the task and domain, the paper currently suffers from insufficiently clear presentation and storytelling. The language, organization, and overall structure of the manuscript require significant polishing and revision. The rebuttal did not fully alleviate these concerns, and the final version should make a stronger effort to improve readability and clarity.

**Audience:**

Yes

**Audience Explanation:**

Researchers, especially engineers, working on robotic navigation and embodied intelligence will likely find this paper interesting and relevant.

**Claims And Evidence:**

Yes

**Claims Explanation:**

The paper is an impressive engineering feat, with rigorous and comprehensive experiments demonstrating the remarkable performance of the proposed models, which outperform state-of-the-art baselines despite the limited technical novelty.

---

> ### Author Response · Authors · 2026-06-10
> **Additional Revisions Following Reviewer SCQw's Recommendations**
>
> # Revisions Addressing Reviewer SCQW's Comments
> We thank the Action Editor and reviewers for their careful evaluation of our revised manuscript. In the previous revision cycle, we made extensive efforts to address all reviewer comments and are pleased that most concerns were satisfactorily resolved. In this camera-ready revision, following the Action Editor's recommendation, we have revisited the remaining concerns raised by Reviewer SCQW and incorporated additional clarifications and revisions throughout the manuscript. The specific changes are summarized below.
>
> ## 1. Methodology Section Organization
>
> **Reviewer Comment:**
> *The methodology section needs major revision in which problem formulation, background, and novelty are strongly intermixed and it is hard for the audience to delineate the novelties from the backgrounds.*
>
> **Response:**
> We thank the reviewer for this feedback. While the contributions are summarized in the Introduction, we agree that the methodology section can be structured more clearly. We have updated the methodology part in the revised draft (please see Section 1). We have rephrased the Introduction section to better motivate the problem and revised the Related Work section to discuss similar and prior approaches. The core contributions are explicitly listed at the end of the Introduction. The problem statement has also been rephrased in Section 3.1 to improve clarity.
>
> ## 2. Novelty of the Transformer and SSM Encoder Models
>
> **Reviewer Comment:**
> *It is not clear in the transformer and SSM encoder model what the novelty of this paper is. It seems the RGB+D encoding network is very common and trivial.*
>
> **Response:**
> While the RGB and depth encoders are based on established architectures, the novelty of our work lies in the design of RGBD fusion within PixNav-style transformer and SSM policies. In particular, we introduce a dedicated depth encoder and a modified fusion strategy for SSM-based policies, resulting in improved representation learning and substantially higher navigation success rates. Pretrained encoder initialization further improves performance compared with transformer-based baselines.
>
> ## 3. Output Heads and Loss Function
>
> **Reviewer Comment:**
> *The output heads need to be elaborated, and the final objective (loss) function is not defined or discussed.*
>
> **Response:**
> The outputs are discussed in the RGB PixNav model. We kindly refer the reviewer to Section 3.1 and Figure 2 where the output heads are described and labelled. We avoided repeating the same lines since the heads remain unchanged across variants, and their learning is ensured through the defined loss terms. The final objective is defined in Equation (6) and discussed in detail in the revised manuscript.
>
> ## 4. Comparison with VLM/VLA and LLM-Based Policies
>
> **Reviewer Comment:**
> *Comparison with VLM/VLA and LLM-based policies are not discussed. Overall, the experimental evaluation is very narrow and limited. The question is why don't we have the SoTA methods compared with the proposal on the embodied task? Is the task different from those in the embodied benchmark evaluation datasets?*
>
> **Response:**
> Existing VLM/VLA approaches, such as Uni-Navid and ImagineNav, primarily address object-goal navigation through action prediction. In contrast, pixel-goal navigation requires reaching an arbitrary target pixel specified within an image, making direct comparison challenging. Nevertheless, to provide additional context, we include a comparison with state-of-the-art VLM-based goal-instance navigation methods in Appendix A.6.
>
> ## 5. Clarification of the Embodied Navigation Task
>
> **Reviewer Comment:**
> *Can you elaborate on if the embodied task is pixel navigation, image navigation, or object navigation?*
>
> **Response:**
> Embodied navigation can take several forms, including image-goal navigation, object-goal navigation, and pixel-goal navigation. In this paper, we focus on pixel-goal navigation, where the agent is asked to navigate to an image goal defined more precisely by a binary mask representing the goal coordinates (see Section 3.1). We additionally evaluate the trained models on zero-shot object-goal navigation (Appendix A.7) and compare against state-of-the-art goal-instance navigation methods (Appendix A.6).
> ## 6. Training and Validation Set Creation
>
> **Reviewer Comment:**
> *We need clarity on how the training and validation sets are created and partitioned.*
>
> **Response:**
> The training and validation set generation pipeline has been expanded in Section 4. We describe how an SPF agent is used to curate ground-truth trajectories while maintaining a balance across different levels of complexity. Dataset partitioning is performed using the training and evaluation splits of the HM3D scenes dataset. Goal points are randomly sampled and are not reused from the original HM3D scene definitions. The revised manuscript provides a more detailed explanation of this process.

---

> > ### Author Response · Authors · 2026-06-10
> > **Additional Revisions Following Reviewer SCQw's Recommendations**
> >
> > ## 7. RGBD Novelty Claim
> >
> > **Reviewer Comment:**
> > *The paper’s novelty claim around RGBD input should be narrowed and made precise. Can you revisit that the claim of this paper is to study end-to-end RGBD fusion within PixNav-style transformer/SSM policies, rather than being the first to consider depth in the broader pixel-navigation literature.*
> >
> > **Response:**
> > We agree with this suggestion and have revised the manuscript accordingly. The revised text now states:
> >
> > > Our claim is not that depth has never been used in embodied navigation, but rather that, to the best of our knowledge, this is the first study to systematically investigate RGBD fusion within PixNav-style pixel-goal navigation policies.
> >
> > ## 8. Performance Difference Between SSM-RGB and SSM-RGBD
> >
> > **Reviewer Comment:**
> > *Can you elaborate why the performance of the "Causal SSM-RGB PixNav" is higher than "Causal SSM-RGBD PixNav" in Table 3?*
> >
> > **Response:**
> > RGB models outperform RGBD models in the basic SSM variants, although both outperform the transformer-based models. Depth provides geometric information, but without an explicit mechanism to determine when depth cues should be emphasized, naïve fusion can introduce ambiguity. We observe that introducing a depth-gating mechanism enables the model to learn how much depth information to use, leading to improved performance. More importantly, depth-aware models demonstrate advantages in harder navigation scenarios where RGB information alone may be insufficient. This discussion has been clarified in the revised manuscript.
> >
> > ## 9. Network and Optimization Hyperparameters
> >
> > **Reviewer Comment:**
> > *The details on the network and optimization hyper-parameters for re-generation purposes are missing.*
> >
> > **Response:**
> > For all trained variants, the network architecture details and optimization hyperparameters are provided in Appendix A.1.

---

> ### Author Response · Authors · 2026-06-10
> **Summary of Revisions Following the Action Editor's Decision**
>
> ## Additional Revisions
>
> In addition to addressing the reviewer comments, we made several improvements to enhance the clarity and presentation of the manuscript. Specifically, we:
>
> - Rephrased the abstract for improved clarity and readability.
> - Added a visual illustration of the PixNav task in the Introduction (Figure 1) to better motivate pixel-guided navigation.
> - Revised all figures throughout the manuscript to improve visibility and readability.
> - Rephrased the problem statement in Section 3.1 to provide a clearer formulation of the task.
> - Explicitly described the dataset preparation process, including the training and validation splits, in Section 4 to improve reproducibility.
>
> These changes complement the revisions made in response to the reviewers' comments and further improve the manuscript's overall readability and presentation quality.